# LNL+K: Enhancing Learning with Noisy Labels Through Noise Source Knowledge Integration

## Abstract

Learning with noisy labels (LNL) aims to train a high-performing model using a noisy dataset. We observe that noise for a given class often comes from a limited set of categories, yet many LNL methods overlook this. For example, an image mislabeled as a cheetah is more likely a leopard than a hippopotamus due to its visual similarity. In fact, we find that many datasets have meta-data information that directly provides potential noise sources. Thus, in this paper, we explore a task we refer to as Learning with Noisy Labels with noise source Knowledge integration (LNL+K), which assumes we have some knowledge about likely source(s) of label noise that we can take advantage of. We find that integrating noise source knowledge boosts performance, even supporting settings where LNL methods typically fail. For example, LNL+K methods are effective on datasets where noise represents the majority of samples, which breaks a critical premise of most methods developed for the LNL task. We also find that LNL+K methods can boost performance even when the noise sources are estimated rather than provided in the meta-data. Our experiments provide several baseline LNL+K methods that integrate noise source knowledge into state-of-the-art LNL models across five diverse datasets and three types of noise, where we report gains of up to 15% compared to the unadapted methods. Critically, we show that LNL methods fail to generalize on some real-world datasets, even when adapted to integrate noise source knowledge, highlighting the importance of directly exploring our LNL+K task.

## 1 Introduction

High-quality labeled data is valuable for training deep neural networks (DNNs), but it's costly and often corrupted in real-world datasets (Krishna et al., 2016; Yan et al., 2014). Learning with Noisy Labels (LNL) addresses this challenge (Natarajan et al., 2013), aiming to learn from noisy training data while achieving strong generalization performance (Arpit et al., 2017; Song et al., 2022). Prior work addresses this task along two main themes: one involves aligning the noisy data classifier with the clean data classifier through estimated noise transitions (Scott, 2015; Liu & Tao, 2015; Yao et al., 2020b; Xia et al., 2019; Zhang et al., 2021; Kye et al., 2022; Cheng et al., 2022), while the other discriminates between noisy and clean samples (Kim et al., 2021; Mirzasoleiman et al., 2020; Wei et al., 2022; Liu et al., 2020; Iscen et al., 2022; Han et al., 2018b; Karim et al., 2022; Li et al., 2020a). The core challenge in both streams of methods centers on distinguishing potential clean and noisy samples. As shown in Fig. 1-a, most existing methods (*e.g.* (Kim et al., 2021; Mirzasoleiman et al., 2020; Wei et al., 2022; Karim et al., 2022)) detect clean samples by finding similar samples within each class. While effective for well-matching samples, it struggles with those near the decision boundary and outliers. A high noise ratio can also lead to estimating noise distribution instead of category distribution, as seen in the 50% noise in the red category in Fig. 1-a. Knowledge of the noise source is a valuable resource in addressing these LNL challenges. However, in most prior work, this knowledge is assumed to be completely unknown and is neglected.

We observe that noise source knowledge already exists or can be estimated in real-world datasets. Labels are rarely uniformly corrupted across all classes, and some classes are more easily to be confused than others (Tanno et al., 2019). For example, visually similar objects are often mislabeled: *e.g.* wolf and coyote (Song et al., 2019), automobiles and trucks (Krizhevsky et al.,

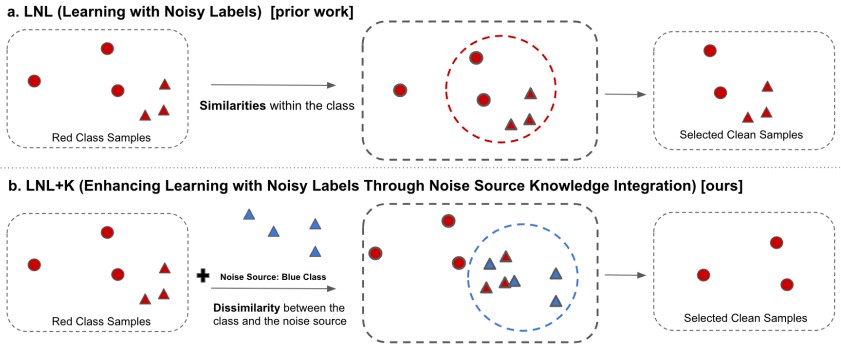

Figure 1: (Best view in color.) **Comparison of a prior work on LNL and our LNL+K task**. **a:** LNL methods select clean samples according to the similarity within the same class (Kim et al., 2021; Mirzasoleiman et al., 2020; Han et al., 2018b; Iscen et al., 2022; 2019; Zhou et al., 2003; Nguyen et al., 2019; Han et al., 2019; Wu et al., 2021), while **b:** LNL+K identify clean samples that are least similar to the noise source. This illustrates a failure case of LNL methods when the noise ratio is high, but where an LNL+K approach can succeed.

2009). Furthermore, in scientific settings, certain categories are intentionally designed to establish causality and can be treated as noise sources during training. For instance, consider the "control" group (Wikipedia contributors, 2022), which represents "do-nothing". In weak-effect classes, the test objects were labeled as having an effect, but visually, they closely resemble the "control" class (*i.e.*, their true label should be "control" ). Importantly, this type of noise ratio can be as high (over 50% (Rohban et al., 2017)). Thus, integrating noise source knowledge offers significant potential, particularly in scientific domains with high noise ratios.

To this end, we explore **L**earning with **N**oisy **L**abels through noise source **K**nowledge integration (LNL+K). In contrast to traditional LNL tasks, we assume that we are given some knowledge about noisy label distribution. *i.e.*, noisy labels tend to originate from specific categories (*e.g.* blue class in Fig. 1). The integration of knowledge about noise sources is helpful in discriminating clean samples in two ways. First, it aids in the identification of hard negative instances. In simpler terms, even if the probability of an instance belonging to a specific category is relatively low, it may still be retained if that probability surpasses the likelihood of it originating from a noise source. For instance, in Fig. 1-b, observe the leftmost red circle; although it appears distant from the red class distribution, it remains preserved as a clean sample in LNL+K because it is even further from the noise source. Second, this integration enables the detection of noisy samples with similar features even at high noise ratios. Specifically, instances with a relatively high probability of belonging to a particular category may not be selected if their likelihood is lower than that of originating from a noise source. For example, in Fig. 1-b, even though the noisy red triangles share similarities with the true circle class, when considering the presence of the noise source yellow class, it becomes evident that these noisy samples are closer to their true label class.

The work most similar to ours is Han et al. (2018a), which introduces a form of noise supervision by removing invalid noise transitions through human cognition. While their paper also aims to leverage some knowledge about the noise source, they focus their exploration on estimating noise transitions to avoid overfitting to noisy labels. In our paper, we update and greatly expand on their initial work, including introducing a unified framework with which we can adapt recent methods from LNL to our LNL+K task (*e.g.*, (Mirzasoleiman et al., 2020; Kim et al., 2021; Wei et al., 2022; Karim et al., 2022)), evaluating on three real world noisy datasets (Chen et al., 2023; Pratapa et al., 2021; Song et al., 2019), and investigating new noise settings designed to reflect applications to scientific datasets. In addition, we also show our LNL+K methods can also take advantage of estimated noise sources using methods from prior work Yao et al. (2020b), effectively bridging the gap between methods that estimate noise transitions to those that focus on identifying clean samples.

In summary, our contributions are:
• We explore an important but overlooked task, termed **LNL+K**: Enhancing **L**earning with **N**oisy **L**abels through noise source **K**nowledge integration. We also design a new noise setting: dominant noise, where noisy samples are the majority of a labeled category distribution.

- We define a unified framework for clean label detection in LNL+K, and explore four baseline methods for LNL+K by adapting LNL methods with noise source knowledge. Additionally, we introduce methods to seamlessly integrate the two distinct streams of LNL approaches: noise estimation and differential training for noisy and clean samples.
- Our experiments show an up to 8% accuracy gain under asymmetric noise and a remarkable 15% performance boost under dominant noise when applied to synthesized datasets using CIFAR-10/CIFAR-100 (Krizhevsky et al., 2009). We also obtain a 2% accuracy gain on CHAMMI-CP (Chen et al., 2023) and 1.5% accuracy gain on BBBC036 (Bray et al., 2016), two real-world noisy datasets for image-based cell profiling (Pratapa et al., 2021), and 1% accuracy gain on Animal10N (Song et al., 2019), a real-world noisy dataset of human-labeled online images.

## 2 RELATED WORK

To tackle the challenge of training with noisy labels, two distinct approaches have emerged: the development of *consistent* and *inconsistent classifiers*. Additionally, some research explores the use of *noise supervision* to aid in learning with noise.

***Classifier-consistent*** methods endeavor to align a classifier trained on noisy data with the optimal classifier, which typically minimizes errors on clean data. The basic idea is that given the noisy class posterior probability $P(\widetilde{Y}|X = x)$ (which can be learned using noisy data) and the transition matrix $T(X = x)$ where $T_{ij}(X = x) = P(\widetilde{Y} = j|Y = i, X = x)$, the clean class posterior probability $P(Y|X = x)$ can be inferred. ($X$ is the input, $Y$ is the true label and $\widetilde{Y}$ is the noisy label). To estimate the transition matrix, some work proposed the use of "anchor points", which are data points that have very high probabilities of belonging to a certain class (Scott, 2015; Liu & Tao, 2015; Menon et al., 2015; Patrini et al., 2017). To avoid using additional clean data, more work focuses on estimating the transition matrix with noisy data (Li et al., 2021; Yao et al., 2020b; Xia et al., 2019; Zhang et al., 2021; Kye et al., 2022; Cheng et al., 2022). Statistically consistent methods train both noisy and clean data indiscriminately but heavily depend on the accuracy of the noise transition matrix, which becomes particularly challenging with high noise ratios. LNL+K can indirectly enhance datasets that require noise source estimation. This can be achieved by combining our proposed task with a method designed to estimate the noise source.

***Classifier-inconsistent*** methods aim to discriminate between clean and noisy labels and handle them differently during the training process. For clean and noisy sample detection, there are mainly loss-based methods that detect noisy samples with high losses (Jiang et al., 2018; Li et al., 2020a; Arazo et al., 2019), and probability-distribution-based approaches that select clean samples with high confidence (Hu et al., 2021; Torkzadehmahani et al., 2022; Nguyen et al., 2019; Tanaka et al., 2018; Li et al., 2022). However, these assumptions may not always hold true, especially with hard negatives and samples along distribution boundaries. Samples selected by these approaches are more likely to be "easy" samples instead of "clean" samples. Feature-based approaches have also been proposed that utilize the input before the softmax layer – high-dimensional features (Mirzasoleiman et al., 2020; Kim et al., 2021), which are less affected by noisy labels (Li et al., 2020b; Yao et al., 2020a; Bai et al., 2021). To differentiate the training of noisy and clean samples, there are methods adjusting the loss function (Wei et al., 2022; Ma et al., 2020; Iscen et al., 2022; Xu et al., 2019; Zhang & Sabuncu, 2018), using regularization techniques (Liu et al., 2020; Xia et al., 2021; Hu et al., 2019), multi-round learning only with selected clean samples (Cordeiro et al., 2023; Shen & Sanghavi, 2019; Wu et al., 2020), and training noisy samples with semi-supervised learning (SSL) techniques (Sohn et al., 2020; Tarvainen & Valpola, 2017; Li et al., 2020a; Karim et al., 2022). To our knowledge, most statistically inconsistent methods often overlook the valuable resource of noise distribution knowledge in the context of LNL. LNL+K makes a unique contribution by utilizing noise source knowledge to detect clean samples within these methods.

***LNL with noise supervision*** methods can be considered precursors to the broader concept of LNL+K (Hendrycks et al., 2018; Li et al., 2017; Yu et al., 2023; Veit et al., 2017; Han et al., 2018a). The small clean dataset is a source of supervision and is employed to achieve an accurate estimation of the noise distribution. However, obtaining human-verified clean datasets is costly and often unavailable. Han et al. (2018a) propose using human cognition of invalid class transitions as "mask" to reduce the burden of transition matrix estimation. However, this approach is constrained to classifier-consistent methods, and the outcomes become unreliable when the noise structure is

misidentified. It's worth noting that previous noise supervision methods often have limitations, such as the requirement for small clean datasets or being applicable only to specific methods. Our LNL+K task has no strict requirement for knowledge to be complete. In essence, the more comprehensive and accurate the knowledge, the more favorable the outcomes. Our motivation lies in the idea that if there exists prior knowledge about a dataset, it is reasonable to utilize it for model training. Even partial or somewhat useful information can offer benefits compared to having no knowledge at all.

## 3 LEARNING WITH NOISY LABELS + KNOWLEDGE (LNL+K)

Learning with Noisy Label Source Knowledge (LNL+K) aims to find the optimal parameters set $\theta^*$ for the classifier $f_\theta$, which is trained on the noisy dataset $D$ **with noise source knowledge** $D_{ns}$ and achieve high accuracy performance on the clean test dataset. In this section, we first introduce the notation we will use, then we will define a unified clean-sample-detection framework in Section 3.1.

Suppose we have a dataset $D = \{(x_i, y_i)_{i=1}^n \in R^d \times K\}$, where $K = \{1, 2, ..., k\}$ is the categorical label for $k$ classes. $(x_i, y_i)$ denotes the $i-th$ example in the dataset, such that $x_i$ is a $d-dimiensional$ input in $R^d$ and $y_i$ is the label. $\{y_i\}_{i=1}^n$ might include noisy labels and the true labels $\{\widetilde{y_i}\}_{i=1}^n$ are unknown. However, we do have some prior knowledge about noisy label sources. Knowledge can take various forms—it can be precise, such as the noise transition matrix obtained through noise modeling methods, or it can be imprecise and incomplete, stemming from human cognition, *e.g.* classes like "cat" and "lynx" are visually similar and are more likely mislabeled with each other (Song et al., 2019). Additionally, knowledge can be derived from the dataset design, *e.g.* "control" class serves as the noise source in scientific datasets. Accordingly, the noise source distribution knowledge $D_{ns}$ can be represented in different ways. One representation is by a probability matrix $P_{k \times k}$, where $P_{ij}$ refers to the probability that a sample in class $i$ is mislabeled as class $j$. Alternatively, it can also be represented using a set of label pairs $LP = \{(i, j) | i, j \in K\}$, where $(i, j)$ refers to the fact that samples in class $i$ are more likely to be mislabeled as class $j$. For the convenience of formulating the following equations, noise source knowledge $D_{c-ns}$ represents the set of noise source labels of category $c$. *I.e.*, $D_{c-ns} = \{i | i \in K \wedge (P_{ic} > 0 \vee (i, c) \in LP)\}$.

### 3.1 A UNIFIED FRAMEWORK FOR CLEAN SAMPLE DETECTION WITH LNL+K

To make our framework general enough to represent different LNL methods, we define a unified logic of clean sample detection. Formally, consider sample $x_i$ with a clean categorical label $c$, *i.e.*,

$$\widetilde{y_i} = c \leftrightarrow y_i = c \wedge p(c|x_i) > \delta, \tag{1}$$

where $p(c|x_i)$ is the probability of sample $x_i$ with label $c$ and $\delta$ is the threshold for the decision. Different methods vary in how they obtain $p(c|x_i)$. For example, as mentioned in related work, loss-based detection uses $Loss(f_\theta(x_i), y_i)$ to estimate $p(c|x_i)$ (Jiang et al., 2018; Li et al., 2020a; Arazo et al., 2019), probability-distribution-based methods use the logits or classification probability score $f_\theta(x_i)$ (Hu et al., 2021; Torkzadehmahani et al., 2022; Nguyen et al., 2019; Tanaka et al., 2018; Li et al., 2022), and feature-based method use $p(c|x_i) = M(x_i, \phi_c)$ (Mirzasoleiman et al., 2020; Kim et al., 2021), where $M$ is a similarity metric and $\phi_c = D(g(X_c))$ is the distribution of features labeled as category $c$, *i.e.*, $X_c = \{x_i | y_i = c\}$, $g(X_c) = \{g(x_i, c) | x_i \in X_c\} \sim \phi_c$, and $g(\cdot)$ is a feature mapping function. The feature-based methods often vary in how they implement their feature mapping $g(\cdot)$ function and similarity distance metric $M$.

LNL+K adds knowledge $D_{ns}$ by comparing $p(c|x_i)$ with $p(c_n|x_i)$, where $c_n$ is the noise source label. When category $c$ has multiple noise source labels, $p(c|x_i)$ should be greater than any of these. In other words, the probability of sample $x_i$ has label $c$ (*i.e.*,$p(c|x_i)$), not only depends on its own value but is decided by the comparison to the noise source labels. For example, the red triangle $x_i$ in Fig. 1 has a high probability of belonging to the red class, *i.e.*, $p(red|x_i) > \delta$, then it is detected as a clean sample in LNL. However, compared to the probability of belonging to the noise source yellow class, $p(yellow|x_i) > p(red|x_i)$, so the red triangle is detected as a noisy sample in LNL+K. To summarize, the propositional logic of LNL+K is:

$$\widetilde{y_i} = c \leftrightarrow y_i = c \wedge p(c|x_i) > Max(\{p(c_n|x_i)|c_n \in D_{c-ns}\}). \tag{2}$$

It's important to clarify that Eq. 2 in LNL+K differs from the conventional LNL approach where a model selects examples with the highest probability for a given class. Two key distinctions are:

- The selected sample's probability may not be the highest. This is particularly beneficial for identifying hard negative samples, such as images with similar backgrounds. While the objects themselves may not exhibit similar features and are not categorized as noise sources, the shared background can cause model confusion during object classification.
- Introducing cross-class comparisons with feature similarity is novel. To our knowledge, existing LNL methods utilizing feature space similarity do not incorporate cross-class comparisons, focusing solely on the likelihood of samples belonging to their designated class (Eq. 1). Although AUM (Pleiss et al., 2020) introduced cross-class comparisons to the non-assigned label with the highest probability, it was limited to logit values and performed relatively poor at high noise levels.

## 3.2 INCORPORATING NOISE SOURCE KNOWLEDGE INTO LNL METHODS

We adapt several recent methods from the LNL literature to support our LNL+K task. Our adaptations enhance the detection of clean samples in *inconsistent-classifier methods*. Once the probability of samples being clean is determined, the remainder of the training process follows the original methods. We provide a summary of each adaptation below, but additional details can be found in Appendix A. Algorithm 1 summarizes our framework's steps offering a unified approach to integrating noise source knowledge through cross-class comparisons. Each base model primarily differs in the function $p(c|x_i)$, which calculates the probability of a sample being clean.

**CRUST**$^{+k}$ adapts CRUST (Mirzasoleiman et al., 2020), which uses the pairwise gradient distance within the class for clean sample detection. A clean sample subset is selected with the most similar gradients clustered together. To estimate the likelihood of a sample label being clean in CRUST$^{+k}$, we mix this sample with all other noise source class samples and apply CRUST to the combined set. If the sample is selected as part of the noise source class cluster, we assume its label is noisy.

**FINE**$^{+k}$ is derived from FINE (Kim et al., 2021), which uses feature eigenvectors for detection. The alignment between a sample and its label class is determined by the cosine distance between the sample's features and the eigenvector of the class feature gram matrix, which serves as the feature representation of that category. FINE then fits a Gaussian Mixture Model (GMM) on the alignment distribution to divide samples into clean and noisy groups - the clean group has a larger mean value, which refers to a better alignment with the category feature representation. The adaptation incorporates the noise source class in alignment calculation. In FINE$^{+k}$, the clean probability is the difference between label-class and noise-source-class alignment. Clean samples have higher alignment differences, while noisy labels have lower values.

**SFT**$^{+k}$ is based on SFT (Wei et al., 2022), which identifies noisy samples by comparing their predictions in the latest few epochs. A sample is detected as noisy if it is classified correctly at the previous epoch but is misclassified in the latest epoch. SFT$^{+k}$ is adapted by restricting the misclassified labels only to noise source labels.

**UNICON**$^{+k}$ is adapted from UNICON (Karim et al., 2022), which estimates the clean probability by using Jensen-Shannon divergence (JSD), a metric for distribution dissimilarity. Disagreement between predicted and one-hot label distributions is utilized, ranging from 0 to 1, with smaller values indicating a higher probability of the label being clean. UNICON$^{+k}$ integrates the noise source knowledge by adding the comparison of JSD with the noise source class. If the sample's predicted distribution aligns more with the noise source, it is considered noisy.

**DualT+X**$^{+k}$ combines noise estimation and noise discrimination methods. DualT (Yao et al., 2020b) is a consistent-classifier method that estimates noise transitions by factorizing the transition matrix into two new matrices that are often easier to estimate compared to the original matrix. Its estimated noise transition matrix can serve as input for any LNL+K method denoted as X$^{+k}$.

## 4 EXPERIMENTS

### 4.1 DATASETS AND EXPERIENTIAL SETTINGS

**Baselines.** In addition to the baseline methods described in Section 3.2, we provide three additional points of comparison. First, *standard training* refers to training on the noisy datasets without any changes (*i.e.*, without using either an LNL or LNL+K method). Second, *oracle* refers to training

---

**Algorithm 1:** Noise Source Integration Algorithm.

---

**Input** : Inputs $X = \{x_i\}_{i=1}^n$, noisy labels $\widetilde{Y} = \{\widetilde{y_i}\}_{i=1}^n$, probability function $p$ in
adaptation-base-method of sample $x$ with label $c$, noise source knowledge $D_{ns}$
**Output:** Probabilities of samples being clean $P(X) = \{p(\widetilde{y_i}|x_i)\}_{i=1}^n$
$P \leftarrow [0] * len(n)$;
**for** $i \leftarrow 1$ **to** $n$ **do**
    $p_i \leftarrow p(\widetilde{y_i}|x_i)$ ; // Get the probability of given label $\widetilde{y_i}$ being clean.
    **for** $c$ *in* $D_{ns}$ **do**
        // Loop through noise sources.
        **if** $p(c|x_i) > p_i$ **then**
            /* If $x_i$ is more likely to belong to the noise source label $c$,
                  then $\widetilde{y_i}$ is considered as the noisy label.         */
            $p_i \leftarrow 0$;
            break;
        **end**
    **end**
    $P[i] \leftarrow p_i$;
**end**

---

with clean labels. Third, we also train with a ground truth transition matrix to provide an upper bound for methods that focus on estimating this matrix (*e.g.*, DualT Yao et al. (2020b)). In the results tables, this method is abbreviated as "GT-T".

### 4.1.1 CIFAR DATASET WITH SYNTHESIZED NOISE

CIFAR-10/CIFAR-100 (Krizhevsky et al., 2009) dataset contains 10/100 classes, with 5000/500 images per class for training and 1000/100 images per class for testing. We applied two synthesized noises with different noise ratios:

- **Asymmetric Noise** simulates real-world scenarios where visually similar objects are more easily to be mislabeled as each other. Noise is generated by corrupting labels specifically for visually similar classes, *e.g.* trucks $\leftrightarrow$ automobiles. Because this type of noise is bi-directional, when the noise ratio exceeds 50%, it becomes difficult for the model to distinguish between high noise ratios with noisy labels and low noise ratios with true labels. Therefore, our experiments focus on two noise ratios: 20% and 40%. See B.1 for the list of confusing pairs. These confusing pairs serve as noise sources in our experiments.

- **Dominant Noise** is a new setting that simulates high-noise ratios in real-world datasets, especially in scientific datasets. We label classes as either "dominant" or "recessive", where samples mislabeled as the "recessive" are likely from the "dominant". In CIFAR-10/CIFAR-100 (Krizhevsky et al., 2009) dataset, we set half categories as different "recessive" classes and the other half categories are different "dominant" classes. Noisy labels are generated by labeling images in "dominant" as "recessive"(*i.e.* images labeled as "dominant" is free of noise and images labeled as "recessive" contain noise). In contrast to symmetric noise, where noisy samples are uniformly distributed across multiple classes, assuming the existence of "dominant" noise source class(es) is more plausible. In addition, the number of clean samples in a class with high symmetric noise is still significantly higher than the number of noisy samples from each class. For example, in the 50% symmetric noise ratio CIFAR-10 setting, where 50% of the noise is uniformly distributed across the other 9 classes, resulting in approximately 6% noise from each class, the number of clean samples in a class still surpasses the number of noisy samples by a factor of 10. While in dominant noise, 50% of the noise is only from the "dominant" class, thus, the class distribution is more likely to be skewed by the noisy labels. Note that this breaks the informative dataset assumption used by prior work (Cheng et al., 2020). See Appendix B.1 for noise composition details. The "dominant" classes serve as noise sources to "recessive" classes in our experiments.

### 4.1.2 REAL-WORLD DATASETS WITH NATURAL NOISE

- **Cell Datasets BBBC036 and CHAMMI-CP** contain single U2OS cell (human bone osteosarcoma) images from the Cell Painting (Bray et al., 2016) datasets, which represent large treatment

Table 1: Asymmetric noise results on CIFAR-10 and CIFAR-100 dataset. (Baselines: DualT (Yao et al., 2020b), CRUST (Mirzasoleiman et al., 2020), FINE (Kim et al., 2021), SFT (Wei et al., 2022) and UNICON (Karim et al., 2022).)The best test accuracy is marked in bold, and the better result between LNL and LNL+K methods is marked with underlined. Relative improvement percentages are marked in red and green. We find knowledge-adapted methods can alter the rankings of the base methods. (*e.g.* SFT and FINE at a noise ratio of 0.4 on CIFAR-100.) See Section 4.2 for discussion.

| | CIFAR-10 | | CIFAR-100 | |
| --- | --- | --- | --- | --- |
| Noise ratio | $0.2 \, {}^+_-$ | $0.4 \, {}^+_-$ | $0.2 \, {}^+_-$ | $0.4 \, {}^+_-$ |
| Standard Training | $91.65 \pm 0.09$ | $91.56 \pm 0.27$ | $72.29 \pm 0.05$ | $70.12 \pm 0.06$ |
| DualT | $92.24 \pm 0.10$ | $66.23 \pm 0.03$ | $53.61 \pm 1.49$ | $52.03 \pm 1.92$ |
| GT-T | $92.51 \pm 0.03$ | $89.68 \pm 0.13$ | $73.88 \pm 0.04$ | $66.61 \pm 0.03$ |
| CRUST | $\underline{91.94 \pm 0.05}$ | $\underline{89.40 \pm 0.03}$ | $60.75 \pm 1.87$ | $59.79 \pm 0.89$ |
| CRUST$^{+k}$ | $89.47 \pm 0.17$ _2.69%_ | $84.96 \pm 0.91$ _4.97%_ | $\underline{62.44 \pm 0.84}$ _2.78%_ | $\underline{61.07 \pm 0.16}$ _2.14%_ |
| FINE | $89.07 \pm 0.03$ | $85.51 \pm 0.18$ | $65.42 \pm 0.11$ | $65.11 \pm 0.11$ |
| FINE$^{+k}$ | $\underline{90.87 \pm 0.04}$ _2.02%_ | $\underline{89.15 \pm 0.26}$ _4.26%_ | $\underline{73.59 \pm 0.12}$ _12.49%_ | $\underline{72.87 \pm 0.11}$ _11.92%_ |
| SFT | $92.67 \pm 0.04$ | $89.77 \pm 0.14$ | $\underline{74.41 \pm 0.05}$ | $69.51 \pm 0.06$ |
| SFT$^{+k}$ | $\underline{\mathbf{93.19 \pm 0.08}}$ _0.56%_ | $\underline{90.55 \pm 0.06}$ _0.87%_ | $74.29 \pm 0.14$ _0.16%_ | $\underline{70.94 \pm 0.13}$ _2.06%_ |
| UNICON | $92.42 \pm 0.04$ | $\underline{\mathbf{91.51 \pm 0.12}}$ | $75.95 \pm 0.04$ | $73.08 \pm 0.07$ |
| UNICON$^{+k}$ | $\underline{92.60 \pm 0.07}$ _0.19%_ | $91.35 \pm 0.24$ _0.17%_ | $\underline{\mathbf{76.87 \pm 0.24}}$ _1.21%_ | $\underline{\mathbf{73.97 \pm 0.11}}$ _1.22%_ |
| Oracle | $93.34 \pm 0.03$ | $92.81 \pm 0.09$ | $74.42 \pm 0.02$ | $73.73 \pm 0.13$ |

screens of chemical and genetic perturbations. Each treatment is tested with multi-well plates and then imaged with the Cell Painting protocol (Bray et al., 2016), which is based on six fluorescent markers captured in five channels. BBBC036[1] sampled single-cell images from 1500 bioactive compounds (treatments) and CHAMMI-CP[2] sampled 7 compounds out of 1500, including "control" group. Our goal is to classify the effects of treatments with cell morphology features trained by the model. A significant challenge is that cells have different degrees of reaction to the treatment, *i.e.*, some treatments are so weak that little difference can be recognized from control features. Thus, the noisy labels in this dataset are those cell images that look like controls (doing-nothing group) but are labeled as treatments. In fact, around 1300 of the 1500 treatments show high feature similarity with the control group (Bray et al., 2016). The true noise ratio is unknown and for those weak treatments, the majority of the cell images might all be noisy. For BBBC036, we reconstructed the cell dataset with 100 treatments, including the "control" treatment. In the case of CHAMMI-CP, we removed three treatments that only appeared in the test set, resulting in four classes: "weak", "medium", "strong" treatments, and "control". The noise source knowledge we applied in these two datasets is that "control" is the noise source to all other classes. Results are reported on these reconstructed datasets and see Appendix B.2, B.3 for the full list.

- **Animall10N** (Song et al., 2019) is a noisy dataset of human-labeled online images. The images were obtained by crawling several online search engines using predefined labels as search keywords. For a clean test set, these images were classified by humans. It contains 5 pairs of confusing animals with a total of 50,000 training images and 5000 testing images. The authors of the dataset noted 5 pairs of classes[3] that can be easily confused. These confusing pairs serve as noise sources in our experiments.

## 4.2 RESULTS

**Asymmetric noise.** Table 1 summarizes the performances in asymmetric noise settings, which shows the advantage of LNL+K in visually similar noise cases. Our adaptation methods consistently outperform the original methods in most noise settings. Importantly, FINE$^{+k}$ demonstrates significant performance improvement, achieving up to an 8% increase in accuracy when compared to the base FINE method on the CIFAR-100 dataset. Additionally, in the case of CIFAR-100 with a 0.4 noise ratio, the base model SFT achieves 9% higher accuracy than FINE. However, with the

---

[1] Available at **https://bbbc.broadinstitute.org/image_sets**

[2] Available at **https://zenodo.org/record/7988357**

[3] Available at **https://dm.kaist.ac.kr/datasets/animal-10n/**

Table 2: Dominant noise results on CIFAR-10 and CIFAR-100 dataset. (Baselines: DualT (Yao et al., 2020b), CRUST (Mirzasoleiman et al., 2020), FINE (Kim et al., 2021), SFT (Wei et al., 2022) and UNICON (Karim et al., 2022).) The best test accuracy is marked in bold, and the better result between LNL and LNL+K methods is marked with underlined. Relative improvement percentages are marked in red and green. We find incorporating source knowledge helps in almost all cases. See Section 4.2 for discussion.

| | CIFAR-10 | | | CIFAR-100 | | |
|---|---|---|---|---|---|---|
| Noise ratio | $0.2\ ^+_-$ | $0.5\ ^+_-$ | $0.8\ ^+_-$ | $0.2^+_-$ | $0.5^+_-$ | $0.8^+_-$ |
| Standard Training | $85.47\ ^{\pm0.52}$ | $85.46\ ^{\pm0.25}$ | $78.99\ ^{\pm0.07}$ | $50.37\ ^{\pm0.45}$ | $41.41\ ^{\pm1.47}$ | $27.03\ ^{\pm0.12}$ |
| DualT | $86.55\ ^{\pm0.06}$ | $83.70\ ^{\pm0.04}$ | $46.96\ ^{\pm0.07}$ | $34.88\ ^{\pm0.11}$ | $27.04\ ^{\pm0.07}$ | $19.94\ ^{\pm0.04}$ |
| GT-T | $88.09\ ^{\pm0.04}$ | $85.24\ ^{\pm0.06}$ | $76.03\ ^{\pm0.04}$ | $59.32\ ^{\pm0.14}$ | $48.39\ ^{\pm0.21}$ | $35.96\ ^{\pm0.04}$ |
| CRUST | $88.21\ ^{\pm0.22}$ | $80.46\ ^{\pm0.17}$ | $65.79\ ^{\pm0.62}$ | $53.48\ ^{\pm0.80}$ | $48.87\ ^{\pm0.31}$ | $35.56\ ^{\pm1.38}$ |
| CRUST$^{+k}$ | $89.53\ ^{\pm0.05}_{1.50\%}$ | $87.19\ ^{\pm0.08}_{8.36\%}$ | $80.54\ ^{\pm0.30}_{22.42\%}$ | $58.69\ ^{\pm0.50}_{9.74\%}$ | $51.56\ ^{\pm0.31}_{5.50\%}$ | $38.07\ ^{\pm2.05}_{7.06\%}$ |
| FINE | $86.23\ ^{\pm0.30}$ | $84.43\ ^{\pm0.38}$ | $75.45\ ^{\pm0.74}$ | $53.68\ ^{\pm1.54}$ | $52.87\ ^{\pm0.98}$ | $39.45\ ^{\pm0.25}$ |
| FINE$^{+k}$ | $88.69\ ^{\pm0.06}_{2.85\%}$ | $88.00\ ^{\pm0.11}_{4.23\%}$ | $80.52\ ^{\pm0.28}_{6.72\%}$ | $57.22\ ^{\pm1.16}_{6.59\%}$ | $54.77\ ^{\pm1.68}_{3.59\%}$ | $42.25\ ^{\pm0.27}_{7.10\%}$ |
| SFT | $89.48\ ^{\pm0.21}$ | $85.43\ ^{\pm0.13}$ | $75.43\ ^{\pm0.12}$ | $51.82\ ^{\pm0.67}$ | $48.21\ ^{\pm1.21}$ | $41.76\ ^{\pm1.34}$ |
| SFT$^{+k}$ | $89.78\ ^{\pm0.03}_{0.34\%}$ | $87.31\ ^{\pm0.15}_{2.20\%}$ | $76.78\ ^{\pm0.38}_{1.79\%}$ | $54.36\ ^{\pm0.48}_{4.90\%}$ | $51.21\ ^{\pm1.14}_{6.22\%}$ | $37.96\ ^{\pm0.05}_{9.10\%}$ |
| UNICON | $90.82\ ^{\pm0.14}$ | $88.43\ ^{\pm0.14}$ | $81.37\ ^{\pm0.43}$ | $63.28\ ^{\pm0.32}$ | $57.92\ ^{\pm0.43}$ | $42.70\ ^{\pm0.50}$ |
| UNICON$^{+k}$ | $\mathbf{90.83}\ ^{\pm0.11}_{0.01\%}$ | $\mathbf{89.21}\ ^{\pm0.42}_{0.88\%}$ | $\mathbf{82.27}\ ^{\pm0.29}_{1.11\%}$ | $\mathbf{66.77}\ ^{\pm0.54}_{5.52\%}$ | $\mathbf{61.55}\ ^{\pm0.13}_{6.27\%}$ | $\mathbf{48.47}\ ^{\pm0.40}_{13.51\%}$ |
| Oracle | $90.85\ ^{\pm0.00}$ | $87.35\ ^{\pm0.00}$ | $82.70\ ^{\pm0.00}$ | $55.85\ ^{\pm0.00}$ | $52.58\ ^{\pm0.00}$ | $44.38\ ^{\pm0.00}$ |

integration of knowledge, FINE$^{+k}$ surpasses the performance of SFT$^{+k}$ with 2%. These results underscore the significance of investigating LNL+K tasks.

**Dominant noise.** Table 2 summarizes performance in dominant noise settings, which shows the advantage of LNL+K moves beyond the noise ratio upper bound limit. Note that in the setting of 80% noise ratio over CIFAR-10 dataset, most methods can not even beat standard training's performance, indicating that noisy samples strongly impact the class distribution, CRUST$^{+k}$, FINE$^{+k}$, and UNICON$^{+k}$ still demonstrate better performance, with UNICON$^{+k}$ coming close to the oracle method's performance and CRUST$^{+k}$ improves the performance by up to 15%.

**Real-world natural noise.** Table 3 reports results for the real-world noisy datasets. We find adaptation methods consistently outperform their base models, underscoring the advantages of knowledge integration without any discernible downsides. For the cell datasets, the presence of high feature similarity between certain treatments and the "control" group can lead to significantly high noise ratios, ultimately strongly influencing the class distribution. BBBC036 and CHAMMI-CP classification tasks are extremely challenging, where only CRUST$^{+k}$ outperforms standard training, boosting top-1 accuracy by 1.5% on BBBC036. Note that our LNL+K methods achieve the best performance across all three datasets. Specifically, CRUST$^{+k}$ improves by 2% on CHAMMI-CP compared to CRUST (Mirzasoleiman et al., 2020), and FINE$^{+k}$ improves by 1% on Animal10N.

We also explored estimating noise source knowledge using DualT (Yao et al., 2020b) with knowledge-adapted methods. Table 4 reports performance, where we find combining DualT with our LNL+K methods boosts performance. Notably, when compared to the original LNL variants from Table 3, our LNL+K models obtain similar or better performance even when estimating noise source knowledge, further validating the importance of our work.

### 4.3 DISCUSSION: KNOWLEDGE ABSORPTION AND FUTURE DIRECTIONS

From the results in Section 4.2, we notice that the accuracy improvements of the adaptation methods vary in different noise settings and methods. We define this different degree of improvement as *knowledge absorption rate*. The values are shown with green/red percentages in the results tables.

**Knowledge absorption rate varies for different methods at the same noise settings.** Considering the unified framework of detecting clean labels in Section 3, $p(c|x_i)$ and $p(c_n|x_i)$ are important factors to *Knowledge absorption rate*. Our baseline methods represent four different methods of

Table 3: Real-world noisy data results. The best test accuracy is marked in bold, and the better result between LNL and LNL+K methods is marked with underlined. Relative improvement percentages are marked in red and green. We find incorporating source knowledge achieves best average accuracy in all datasets. See Section 4.2 for discussion.

| | CHAMMI-CP $^+_-$ | BBBC036 $^+_-$ | Animal10N $^+_-$ |
|---|---|---|---|
| Standard Training | $78.87 \pm 0.45$ | $63.49 \pm 0.62$ | $80.32 \pm 0.20$ |
| DualT (Yao et al., 2020b) | $79.33 \pm 0.17$ | $61.54 \pm 0.61$ | $81.14 \pm 0.28$ |
| CRUST (Mirzasoleiman et al., 2020) | $78.02 \pm 0.31$ | $63.06 \pm 0.65$ | $\underline{81.88} \pm 0.13$ |
| CRUST$^{+k}$ | $\underline{\mathbf{79.81}} \pm 0.56$ ₂.₂₉% | $\underline{\mathbf{65.07}} \pm 0.71$ ₃.₁₉% | $81.74 \pm 0.08$ ₀.₁₇% |
| FINE (Kim et al., 2021) | $\underline{67.27} \pm 0.82$ | $56.80 \pm 0.87$ | $81.15 \pm 0.11$ |
| FINE$^{+k}$ | $67.02 \pm 0.73$ ₀.₃₇% | $\underline{57.01} \pm 0.40$ ₀.₃₇% | $\underline{82.27} \pm 0.10$ ₁.₃₈% |
| SFT (Wei et al., 2022) | $76.08 \pm 0.25$ | $51.71 \pm 0.82$ | $82.24 \pm 0.10$ |
| SFT$^{+k}$ | $\underline{77.75} \pm 0.42$ ₂.₂₀% | $\underline{59.18} \pm 1.33$ ₁₄.₄₅% | $\underline{82.88} \pm 0.18$ ₀.₇₈% |
| UNICON (Karim et al., 2022) | $\underline{71.45} \pm 0.03$ | $33.98 \pm 1.03$ | $87.76 \pm 0.06$ |
| UNICON$^{+k}$ | $71.04 \pm 0.14$ ₀.₅₇% | $\underline{42.17} \pm 0.31$ ₂₄.₁₀% | $\mathbf{88.28} \pm 0.29$ ₀.₅₉% |

| | accuracy |
|---|---|
| DualT (Yao et al., 2020b) | $81.14 \pm 0.28$ |
| DualT+CRUST$^{+k}$ | $80.82 \pm 0.06$ |
| DualT+FINE$^{+k}$ | $81.84 \pm 0.10$ |
| DualT+SFT$^{+k}$ | $81.66 \pm 0.20$ |
| DualT+UNICON$^{+k}$ | $\mathbf{88.42} \pm 0.74$ |

Table 4: Results of combining consistent and inconsistent algorithms with knowledge on Animal10N dataset. We find utilizing noise estimation from a consistent algorithm can boost the performance. See Section 4.2 for discussion.

estimating $p(c|x_i)$. The results conclude that noise source knowledge might be more helpful to the feature-based clean sample detection methods in high noise ratios. CRUST$^{+k}$ has better performance than FINE$^{+k}$ on high noise ratios in the cell dataset. One possible explanation for this is that $p(c|x_i)$ for FINE$^{+k}$ depends on the category feature distribution while CRUST+K focuses on the feature of a single sample and aims to find the subset with minimum gradient distance sum. In other words, when the noise ratio is high, category feature distribution in FINE$^{+k}$ might be skewed while CRUST$^{+k}$ is less affected by finding the optimal cluster. *Knowledge absorption rate* indicates how well an LNL method can transfer to the LNL+K task with noise distribution knowledge, exploring ways to enhance the transferability of LNL methods and optimizing this value are important areas for further investigation.

**Limitations.** Our exploration of LNL+K has primarily focused on the assumption of closed-set noise. A potential future research direction could involve investigating LNL+K in the context of open-set noise, which is more prevalent in real-world datasets, particularly those from web crawling. Furthermore, it's worth noting that the noise source knowledge we've employed has been restricted to category-dependent information.

## 5 CONCLUSION

This paper introduces a new task, LNL+K, which leverages noise source distribution knowledge when learning with noisy labels. This knowledge is not only beneficial to distinguish clean samples that are ambiguous or out-of-distribution but also necessary when the noise ratio is so high that the noisy samples dominate the class distribution. Instead of comparing the "similarity" of the samples within the same class to detect the clean ones, LNL+K utilizes the "dissimilarity" between the sample and the noise source for detection. We provide a unified framework of clean sample detection for LNL+K which we use to adapt state-of-the-art LNL methods, CRUST$^{+k}$, FINE$^{+k}$, SFT$^{+k}$, and UNICON$^{+k}$ to our task. To create a more realistic simulation of high-noise-ratio settings, we introduce a novel noise setting called "dominant noise." Results show LNL+K methods have up to 8% accuracy gains over asymmetric noise and up to 15% accuracy gains in the dominant noise setting. Finally, we discuss "knowledge absorption", which notes the ranking of LNL methods to our task varies from their LNL performance, indicating that direct investigation of LNL+K is necessary.

## 6 CODE OF ETHICS STATEMENT

The improved results on the cell dataset imply that our work opens the door to LNL in scientific settings. At the same time, our work will have a social impact on domain experts, who can avoid some labor-intensive jobs such as correcting labels of medical images. However, we are also aware that LNL can enable bad actors to train a high-performing model as well.

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

# A  LNL+K BASELINE METHODS

## A.1  CRUST$^{+k}$

The key idea of CRUST (Mirzasoleiman et al., 2020) is from the neural network Jacobian matrix containing all its first-order partial derivatives. It is proved in their work that the neural network has a low-rank Jacobian matrix for clean samples. In other words, data points with clean labels in the same class often have similar gradients clustered closely together. CRUST is a feature-based method and this approach can be summarized with settings in Section 3.1. The feature used for selection is the pairwise gradient distance within the class: $g(X_c) = \{d_{x_i x_j}(\mathcal{W}) | x_i, x_j \in X_c\}$, where $d_{x_i x_j}(\mathcal{W}) = \|\nabla L(\mathcal{W}, x_i) - \nabla L(\mathcal{W}, x_j)\|_2$, $\mathcal{W}$ is the network parameters and $L(\mathcal{W}, x_i) = \frac{1}{2}\sum_{x_i \in D}(y_i - f_\theta(\mathcal{W}, x_i))^2$. CRUST needs an additional parameter $\beta$ to control the size of the clean selection set $X'_c$. Given $\beta$, the sample $x_i$ is selected as clean if $\|X'_c\| = \beta$ ($\|X'_c\|$ is the size of set $X'_c$) and $x_i \in X'_c$, where $\sum g(X'_c)$ has the minimum value. *i.e.*, the selected clean subset $X'_c$ has the most similar gradients clustered together. Thus, we can summarize the similarity metric $M$ for $p(c|x_i)$ as:

$$M(x_i, \phi_c, \beta) = 1 \leftrightarrow \exists X'_c \subset X_c \wedge \|X'_c\| = \beta,$$
$$s.t.\ x_i \in X'_c \wedge (\forall \|X''_c\| = \beta \wedge X''_c \subset X_c, \sum g(X'_c) \leq \sum g(X''_c)), \tag{3}$$

otherwise $M(x_i, \phi_c, \beta) = 0$. Thus, we can get the propositional logic of CRUST:

$$\widetilde{y_i} = c \leftrightarrow y_i = c \wedge p(c|x_i) = 1 \leftrightarrow M(x_i, \phi_{y_i}, \beta) = 1. \tag{4}$$

To adapt CRUST to CRUST$^{+k}$ with noise source distribution knowledge. from Eq.2, we have

$$y_i = c \wedge \widetilde{y_i} \neq c \leftrightarrow p(c|x_i) \leq Max(\{p(c_n|x_i) | c_n \in D_{c-ns}\})$$
$$\leftrightarrow \exists c_n \in D_{c-ns}\ s.t.\ p(c_n|x_i) \geq p(c|x_i) \leftrightarrow \exists c_n \in D_{c-ns}\ s.t.\ p(c_n|x_i) = 1. \tag{5}$$

To get $p(c_n|x_i)$, we first mix $x_i$ with all the samples in $X_{c_n}$, *i.e.*, $X_{c_n+} = \{x_i\} \cup X_{c_n}$. Then apply CRUST on this mix set, *i.e.*, calculate the loss towards label $c_n$ and select the clean subset $X'_{c_n+}$. if $x_i \in X'_{c_n+}$, then $p(c_n|x_i) = 1$. Here is the formulation of CRUST$^{+k}$, we modify $L(\mathcal{W}, x_i)$ to $L(\mathcal{W}, x_i, c) = \frac{1}{2}\sum_{x_i \in D}(c - f_\theta(\mathcal{W}, x_i))^2$, where we calculate the loss to any certain categories, not limited to the loss towards the label. Similarly, we have $d_{x_i x_j}(\mathbf{W}, c) = \|\nabla L(\mathcal{W}, x_i, c) - \nabla L(\mathcal{W}, x_j, c)\|_2$, $g(X_{c_n+}, c_n) = \{d_{x_i x_j}(\mathbf{W}, c_n) | x_i, x_j \in X_{c_n+}\}$. We use $\gamma$ to represent the subset size of $X_{c+c_n}$, which is decided by $\beta$ and noise source distribution. Finally, we get the similarity metric $M(x_i, \phi_{c_n+}, \gamma)$ as:

$$M(x_i, \phi_{c_n+}, \gamma) = 1 \leftrightarrow \exists X'_{c_n+} \subset X_{c_n+} \wedge \|X'_{c_n+}\| = \gamma,$$
$$s.t.\ x_i \in X'_{c_n+} \wedge (\forall \|X''_{c_n+}\| = \gamma \wedge X''_{c_n+} \subset X_{c_n+}, \sum g(X'_{c_n+}, c_n) \leq \sum g(X''_{c_n+}, c_n)), \tag{6}$$

otherwise $M(x_i, \phi_{c_n+}, \gamma) = 0$. Combining Eq.2, Eq.4, and Eq.6, $p(c|x_i)$ of CRUST$^{+k}$ method is:

$$\widetilde{y_i} = c \leftrightarrow y_i = c \wedge (\forall c_n \in D_{c-ns}, p(c_n|x_i) < p(c|x_i))$$
$$\leftrightarrow y_i = c \wedge (\forall c_n \in D_{c-ns}, p(c_n|x_i) = 0) \leftrightarrow y_i = c \wedge (\forall c_n \in D_{c-ns}, M(x_i, \phi_{c_n+}, \gamma) = 0). \tag{7}$$

## A.2  FINE$^{+k}$

Filtering Noisy instances via their Eigenvectors(*FINE*) (Kim et al., 2021) selects clean samples with the feature-based method. Let $f_{\theta*}(x_i)$ be the feature extractor output and $\Sigma_c$ be the gram

matrix of all features labeled as category $c$. The alignment is defined as the cosine distance between feature $\overrightarrow{f_{\theta^*}(x_i)}$ and $\overrightarrow{c}$, which is the eigenvector of the $\Sigma_c$ and can be treated as the feature representation of category $c$. FINE fits a Gaussian Mixture Model (GMM) on the alignment distribution to divide samples to clean and noisy groups - the clean group has a larger mean value, which refers to a better alignment with the category feature representation. In summary, feature mapping function $g(x_i, c) = < \overrightarrow{f_{\theta^*}(x_i)}, \overrightarrow{c} >$, and mixture of Gaussian distributions $\phi_c = \mathcal{N}_{clean} + \mathcal{N}_{noisy} = \mathcal{N}(\mu_{g(X_{c-clean})}, \sigma_{g(X_{c-clean})}) + \mathcal{N}(\mu_{g(X_{c-noisy})}, \sigma_{g(X_{c-noisy})})$, where $\mu_{g(X_{c-clean})} > \mu_{g(X_{c-noisy})}$. The similarity metric

$$M(x_i, \phi_c) = \begin{cases} 1 & : & \mathcal{N}_{clean}(g(x_i, c)) > \mathcal{N}_{noisy}(g(x_i, c)) \\ 0 & : & \mathcal{N}_{clean}(g(x_i, c)) \leq \mathcal{N}_{noisy}(g(x_i, c)). \end{cases} \tag{8}$$

Thus, we have

$$\widetilde{y}_i = c \leftrightarrow y_i = c \wedge p(c|x_i) = 1 \leftrightarrow M(x_i, \phi_{y_i}) = 1. \tag{9}$$

Next, we show our design of FINE$^{+k}$ with noise source distribution knowledge. The key difference between FINE and FINE$^{+k}$ is that we use the alignment score of the noise source class. For a formal description of FINE$^{+k}$, We define $g_k(x_i, c, c_n) = g(x_i, c) - g(x_i, c_n)$. Similar to FINE, FINE$^{+k}$ fits a GMM on $g_k(X_c, c, c_n)$, so we have $g_k(X_c, c, c_n) \sim \phi_{k-\{c+c_n\}} = \mathcal{N}_{close-c} + \mathcal{N}_{close-c_n}$, where $\mu_{close-c} > \mu_{close-c_n}$. This can be interpreted in the following way: Samples aligning better with category $c$ should have larger $g(x_i, c)$ values and smaller $g(x_i, c_n)$ values according to the assumption, thus the greater the $g_k(x_i, c, c_n)$, the closer to category $c$, vice versa, the smaller the $g_k(x_i, c, c_n)$, the closer to category $c_n$. Then we have

$$M(x_i, \phi_{k-\{c+c_n\}}) = \begin{cases} 1 & : & \mathcal{N}_{close-c}(g_k(x_i, c, c_n)) > \mathcal{N}_{close-c_n}(g_k(x_i, c, c_n)) \\ 0 & : & \mathcal{N}_{close-c}(g_k(x_i, c, c_n)) \leq \mathcal{N}_{close-c_n}(g_k(x_i, c, c_n)). \end{cases} \tag{10}$$

By combining with Eq.2, we have

$$\begin{aligned} \widetilde{y}_i = c &\leftrightarrow y_i = c \wedge (\forall c_n \in D_{c-ns}, p(c|x_i) > p(c_n|x_i)) \\ &\leftrightarrow y_i = c \wedge (\forall c_n \in D_{c-ns}, M(x_i, \phi_{k-\{c+c_n\}}) = 1). \end{aligned} \tag{11}$$

## A.3 SFT$^{+k}$

SFT (Wei et al., 2022) detects noisy samples according to predictions stored in a memory bank $\mathcal{M}$. $\mathcal{M}$ contains the last $T$ epochs' predictions of each sample. A sample $x_i$ is detected as noisy if a fluctuation event occurs, *i.e.*, the sample classified correctly at the previous epoch $t_1$ is misclassified at $t_2$, where $t_1 < t_2$. The occurrence of the fluctuation event can be formulated as $fluctuation(x_i, y_i) = 1$, otherwise $fluctuation(x_i, y_i) = 0$ *i.e.*,

$$\begin{aligned} fluctuation(x_i, y_i) = 1 &\leftrightarrow \exists t_1, t_2 \in \{t - T, \cdots, T\} \wedge t_1 < t_2 \\ &s.t.\ f_\theta(x_i)^{t_1} = y_i \wedge f_\theta(x_i)^{t_2} \neq y_i, \end{aligned} \tag{12}$$

where $f_\theta(x_i)^{t_1}$ represents the prediction of $x_i$ at epoch $t_1$. SFT is a probability-distribution-based approach and can fit our probabilistic model as follows. The propositional logic of SFT is,

$$p(c|x_i) = \begin{cases} 1 & : & y_i = c \wedge fluctuation(x_i, y_i) = 0 \\ 0 & : & otherwise. \end{cases} \tag{13}$$

*I.e.*, SFT$^{+k}$ applies the noise source distribution knowledge to SFT by stricting the constraints of fluctuation. The fluctuation events only occur when the previous correct prediction is misclassified as the noise source label. Thus, we define SFT$^{+k}$ fluctuation as,

$$\begin{aligned} fluctuation(x_i, y_i, D_{y_i-ns}) = 1 &\leftrightarrow \exists c_n \in D_{y_i-ns}, \exists t_1, t_2 \in \{t - T, \cdots, T\} \wedge t_1 < t_2, \\ &s.t.\ f_\theta(x_i)^{t_1} = y_i \wedge f_\theta(x_i)^{t_2} = c_n. \end{aligned} \tag{14}$$

Combining Eq. 2, Eq. 13 and Eq. 14, SFT$^{+k}$ detects $x_i$ with clean label $y_i = \widetilde{y}_i = c$ with $p(c|x_i)$ as:

$$\begin{aligned} \widetilde{y}_i = c &\leftrightarrow y_i = c \wedge p(c|x_i) > Max(\{p(c_n|x_i)|c_n \in D_{c-ns}\}) \\ &\leftrightarrow y_i = c \wedge p(c|x_i) = 1 \leftrightarrow y_i = c \wedge fluctuation(x_i, y_i, D_{y_i-ns}) = 0. \end{aligned} \tag{15}$$

## A.4 UNICON$^{+k}$

UNICON (Karim et al., 2022) estimate the clean probability by using Jensen-Shannon divergence (JSD) $d_i$, which is a measure of distribution disagreement. JSD is defined by KLD, which is the Kullback-Leibler divergence function. We follow the same JSD definition as UNICON in the adaptation method. Given the predicted probability $p_i$ and label $y_i$, $d_i = JSD(y_i, p_i)$. The value of $d_i$ ranges from 0 to 1 and the smaller the $d_i$ is, the higher the probability of $y_i$ being clean. A cutoff value $d_{cutoff}$ is used to select clean samples. To summarize, the propositional logic of UNICON is,

$$p(c|x_i) = \begin{cases} 1 - JSD(x_i, y_i) & : & y_i = c \wedge JSD(x_i, y_i) < d_{cutoff} \\ 0 & : & otherwise. \end{cases} \tag{16}$$

Then noise source knowledge is integrated with our unified framework:

$$\begin{aligned} \widetilde{y_i} = c \leftrightarrow y_i = c \wedge p(c|x_i) > Max(\{p(c_n|x_i)|c_n \in D_{c-ns}\}) \\ \leftrightarrow y_i = c \wedge (\forall c_n \in D_{c-ns}, JSD(x_i, y_i) < JSD(x_i, c_n)). \end{aligned} \tag{17}$$

## B DATASETS

### B.1 CIFAR DATASET WITH SYNTHESIZED NOISE

**Asymmetric noise.** Labels are corrupted to visually similar classes. Pair $(C_1, C_2)$ represents the samples in class $C_1$ are possibly mislabeled as $C_2$. Noise ratios in the experiments are only the noise ratio in class $C_1$, *i.e.* not the overall noise ratio. Here are the class pairs of CIFAR-10 and CIFAR-100 for asymmetric noise. **CIFAR-10** (trucks, automobiles), (cat, dog), (horse, deer). **CIFAR-100** (beaver, otter), (aquarium fish, flatfish), (poppies, roses), (bottles, cans), (apples, pears), (chair, couch), (bee, beetle), (lion, tiger), (crab, spider), (rabbit, squirrel), (maple, oak), (bicycle, motorcycle).

**Dominant noise** There are "recessive" and "dominant" classes in dominant noise. For CIFAR-10, category index of the last 5 are "recessive" classes and the first five are "dominant" classes. In other words, category index 6-10 samples might be mislabeled as label index 1-5. Different numbers of samples are mixed for different noise ratios so that the dataset is still balanced after mislabeling. Table 5 shows the number of samples per category for each noise ratio.

Table 5: Sample composition for CIFAR-10/CIFAR-100 dominant noise

| CIFAR-10 Dominant Noise | | | |
|---|---|---|---|
| Noise ratio | 0.2 | 0.5 | 0.8 |
| Dominant class | 2000 | 1250 | 500 |
| Recessive class | 3000 | 3750 | 4500 |
| CIFAR-100 Dominant Noise | | | |
| Noise ratio | 0.2 | 0.5 | 0.8 |
| Dominant class | 200 | 125 | 50 |
| Recessive class | 300 | 375 | 450 |

### B.2 CELL DATASET BBBC036

For our experiments we subsampled 100 treatments to evaluate natural noise. Table 6 shows the treatment list. ("NA" refers to the control group, *i.e.* no treatment group.)

### B.3 CELL DATASET CHAMMI-CP

Three compounds with a "control" group are selected for our experiments: BRD-A29260609 (weak reaction), BRD-K04185004 (medium reaction) and BRD-K21680192 (strong reaction).

Table 6: Treatments used from the BBBC036 dataset

| NA | BRD-K88090157 | BRD-K38436528 | BRD-K07691486 | BRD-K97530723 |
|---|---|---|---|---|
| BRD-A32505112 | BRD-K21853356 | BRD-K96809896 | BRD-A82590476 | BRD-A95939040 |
| BRD-A53952395 | BRD-A64125466 | BRD-A99177642 | BRD-K90574421 | BRD-K07507905 |
| BRD-K62221994 | BRD-K62810658 | BRD-K47150025 | BRD-K17705806 | BRD-K85015012 |
| BRD-K37865504 | BRD-A52660433 | BRD-K66898851 | BRD-K15025317 | BRD-K37392901 |
| BRD-K91370081 | BRD-K39484304 | BRD-K03842655 | BRD-K76840893 | BRD-K62289640 |
| BRD-K14618467 | BRD-K52313696 | BRD-K43744935 | BRD-K86727142 | BRD-K21680192 |
| BRD-K06426971 | BRD-K24132293 | BRD-K68143200 | BRD-K08554278 | BRD-K78122587 |
| BRD-A47513740 | BRD-K18619710 | BRD-A67552019 | BRD-K17140735 | BRD-K30867024 |
| BRD-K36007650 | BRD-K51318897 | BRD-K90382497 | BRD-K00259736 | BRD-K95435023 |
| BRD-K52075040 | BRD-K03642198 | BRD-K47278471 | BRD-K17896185 | BRD-K95603879 |
| BRD-A70649075 | BRD-K02407574 | BRD-A90462498 | BRD-K67860401 | BRD-A64485570 |
| BRD-K88429204 | BRD-A49046702 | BRD-K50841342 | BRD-K35960502 | BRD-K77171813 |
| BRD-K54095730 | BRD-K93754473 | BRD-K22134346 | BRD-K72703948 | BRD-K31342827 |
| BRD-K31542390 | BRD-K18250272 | BRD-K00141480 | BRD-K37991163 | BRD-K13533483 |
| BRD-K67439147 | BRD-A91008255 | BRD-K39187410 | BRD-K26997899 | BRD-K89732114 |
| BRD-K50135270 | BRD-K95237249 | BRD-K44849676 | BRD-K20742498 | BRD-K31912990 |
| BRD-K96799727 | BRD-K09255212 | BRD-A89947015 | BRD-K78364995 | BRD-K49294207 |
| BRD-K08316444 | BRD-K89930444 | BRD-K50398167 | BRD-K47936004 | BRD-A72711497 |
| BRD-A97104540 | BRD-A50737080 | BRD-K80970344 | BRD-K50464341 | BRD-K97399794 |

Table 7: Hyperparameters for each dataset.

|  | learning rate | warm-up epochs |
|---|---|---|
| CIFAR-10/CIFAR-100 | 1e-2 | 40 |
| BBBC036 | 2e-4 | 10 |
| CHAMMI-CP | 2e-4 | 5 |
| Animal10N | 5e-3 | 3 |

## C  MODEL

We used a pre-trained ResNet34 on CIFAR-10/CIFAR-100 and Animal10N datasets for all approaches (He et al., 2016) (UNICON trains on two networks (Karim et al., 2022)). For experiments on BBBC036 we used an Efficient B0 for all methods (Tan & Le, 2019) and all methods used ConvNet for CHAMMI-CP dataset (Liu et al., 2022). To support the 5 channel images, we replaced the first convolutional layer in the network to support the new image dimensions.

## D  HYPERPARAMETERS

For a fair comparison, we use the same hyperparameter settings as in prior work (Mirzasoleiman et al., 2020; Kim et al., 2021; Wei et al., 2022; Karim et al., 2022) for CIFAR-10/CIFAR-100 datasets. Hyperparameters of the cell dataset BBBC036 were set via grid search using the validation set. All the experiments use the same batch size of 128. "fl-ratio" of CRUST and CRUST$^{+k}$, which controls the size of selected clean samples is set as the same as the noise ratio in synthesized noise and set as 0.6 in cell dataset BBBC036 and CHAMMI-CP, 0.9 in Animal10N. All the other hyperparameters for each dataset are summarized in Table 7.

