# OpenReview forum: "LNL+K: Enhancing Learning with Noisy Labels Through Noise Source Knowledge Integration"
_ICLR.cc/2024/Conference — ICLR 2024 Conference Withdrawn Submission_

### Official Review · Reviewer_D5C9 · 2023-10-21

**Soundness:** 1 poor
**Presentation:** 3 good
**Contribution:** 1 poor
**Rating:** 1
**Confidence:** 4

**Summary:**

This paper presents a solution to enhance learning with noisy labels using the knowledge of noise source. It presents a few extensions from current solutions to add the noise source knowledge.  Basically, if the sample is more likely to belong to a noise source, then the label can be considered a noisy label. Experimental results show the method works to some extent.

**Strengths:**

The paper presents a simple and straightforward method to detect noisy label under a very strong and unrealistic assumption. Multiple datasets are evaluated to conduct the experiments to show the idea work.

**Weaknesses:**

There are several major limitations of the paper.

(1) The novelty of the paper is very limited. The idea is established under a very strong assumption that the noise source is known. Moreover, mathematically, the method is also unjustified.

(2) The depth of the paper is not strong enough. There is no theory to support the claims. The method also looks ad-hoc. The level of depth is too far away from ICLR paper.

(3) The paper is also not well-written in the sense that the motivation of the paper is not clear.

**Questions:**

In addition to very limited novelty and not enough technical depth, it is also unclear that how is the probability that labels is clean computed and how the noise source is being identified.

For instance, the noise label created by different people with different levels of experience can be mixed without the knowledge of source. It is a more general case. It is important to solve a more general more instead of an edge case.

---

> ### Author Response · Authors · 2023-11-14
> **Reply (part 1)**
>
> Thank you for your efforts in reviewing our paper, and we will use your comments to improve our paper.  We have answered your questions below, and asked for additional clarifications for questions that were unclear to us.
> >(1) The novelty of the paper is very limited. The idea is established under a very strong assumption that the noise source is known.
>
> We will begin by noting that the reviewer's comment provides no justification for their assertions.  We are happy to discuss these questions, and will give a general response, but we ask that the reviewer provide comparisons to existing work in order to justify their claims.
>
> That said, we will begin our response by separating two concepts that the reviewer appears to be conflating, namely that of novelty and impact.  Novelty typically refers to offering a new idea not previously explored.  In our paper, we do indeed propose exploring cases where the noise source is known.  While this is a simple idea, this has not been rigorously explored previously.  In fact, we provide dozens of examples of papers that explored LNL tasks, but did not explore our setting.  The closest work to ours, Han et al, 2018a, also makes a similar argument to ours, but did so in a limited setting that only considered classifier-consistent methods that are unreliable when there are errors in the noise sources.  However, our paper provides a more thorough introduction to this topic including proposing a framework for addressing this task as well as experiments on several real-world datasets.  Of particular note are our cell datasets, which have additional sources of noise that we are not given (discussed further in our response to Reviewer 3qaZ), yet our approach is still able to boost performance in this challenging and realistic setting.
>
> Now, one could also consider the impact as well, which is affected by the assumptions that we make.  Our primary goal is to better support datasets where noise source knowledge is known.  As our experiments in Table 3 demonstrate, existing LNL methods often do not generalize to some settings, such as scientific datasets like CHAMMI-CP and BBBC036 where the noise source knowledge is always known.  Specifically, no method we explore was able to outperform standard training on both datasets.  In fact, the more recent methods actually performed worse than some of the simpler approaches.  This is likely due, in part, to the fact that these datasets are very challenging, with many types of noise and sometimes having high-ratios of noise.  However, as our experiments show, this is where having noise source knowledge is key: as they always perform on par or better than the adaptations that did not provide this information.
>
> Finally, we note that we did also explore settings where the noise source is not known and must be estimated in Table 4.  Specifically, we found that using estimated noise combined with our LNL+K adaptations boosted performance over the methods proposed by prior work.  As such, our work not only explored a setting that prior work failed to generalize to (i.e., they performed worse than standard training), but we also demonstrated that exploring our setting could provide benefits to other datasets where noise source is not known.  Thus, one can only conclude that our work is both novel and makes a significant contribution over the existing literature.
>
> >Moreover, mathematically, the method is also unjustified.
>
> First, we begin by noting that our primary contribution is a study in a largely unexplored topic, namely, where noise source knowledge is known.  Second, we mathematically define this problem thoroughly as well as the guiding principles that we use to adapt existing work to our problem in Section 3.1.  This would seemingly directly contradict the reviewer’s comment, so we request for justification as to why they feel this is insufficient or what questions the reviewer had so that we could use these comments to improve our paper.

---

> > ### Author Response · Authors · 2023-11-14
> > **Reply (part 2)**
> >
> > >(2) The depth of the paper is not strong enough. There is no theory to support the claims. The method also looks ad-hoc. The level of depth is too far away from ICLR paper.
> >
> > We note that our primary contribution is identifying a key problem not well explored in prior work, namely, exploring settings in learning with noisy labels where the source of noise is known.  To help support our claims that this is an important problem that cannot be solved with methods from prior work, we provide experiments in Table 3 on real-world datasets that satisfy our assumptions.  In these experiments, we both verify that (1) prior work does not support these settings (as methods from prior work perform worse than standard training) and, (2) by adapting these methods using our framework outlined in Section 3.1, we could boost performance and outperform standard training.  We also provide a detailed discussion that compares to both prior work (in the related work section) and how we adapt the methods of prior work to our task (summarized in Section 3.1, but far more details in Appendix A).  As such, we find that our paper well supports our claims, and would request the reviewer to clarify exactly what components they feel could use improvement.
> >
> > >(3) The paper is also not well-written in the sense that the motivation of the paper is not clear.
> >
> > Our primary contribution is providing a rigorous and often overlooked task we refer to as Learning with Noisy Labels through noise source Knowledge Integration (LNL+K).  Prior work has primarily focused on settings where the noise sources for a dataset our unknown, but we provide several examples where this assumption made by prior work is not valid.  For example, in scientific settings, such as our cell datasets, there is typically a “control/do nothing group,” which we use as the noise source knowledge for our experiments.  However, we also show that in settings where these noise sources can be estimated, methods designed for our task can also boost performance (Table 4).
> >
> > In summary, we identified a key deficiency in prior work surrounding the assumption that noise source knowledge is typically unknown, but find there are many cases where this is known, and provide the first in-depth study of the topic.

---

> > > ### Author Response · Authors · 2023-11-14
> > > **Reply (part 3)**
> > >
> > > >In addition to very limited novelty and not enough technical depth, it is also unclear that how is the probability that labels is clean computed and how the noise source is being identified.
> > > For instance, the noise label created by different people with different levels of experience can be mixed without the knowledge of source. It is a more general case. It is important to solve a more general more instead of an edge case.
> > >
> > > First, we begin by noting that this argues that only some types of problems are important to solve.  While we generally agree with this statement, the argument that the reviewer makes here is by no means justified.  In particular, if we were to consider the cell data we evaluate on as an example.  This is a case where (A) no annotators are used, (B) the noise source is always known, and (C) current methods do worse than standard training (shown in Table 3).  In other words, this is not an “edge case,” but rather the standard in these datasets, and current methods fail. Additionally, these settings are vital to explore as they have high impact applications.  For example, the cell painting datasets we evaluate on can be directly applied in evaluating drug discovery experiments. In other words, boosting performance on our task can help lead to obtaining new treatments, improving healthcare outcomes, which we can agree is a very important task.
> > >
> > > Second, the reviewer claims that label noise is not always known.  While this is true, we note that prior work has found this typically can be estimated.  This is due to the fact that people generally make similar mistakes, for example, visually similar objects are more often mislabeled than others (Tanno et al., 2019, Song et al., 2019).  As such, it is rare that the noise does not follow a particular pattern (Tanno et al., 2019).  Thus, as demonstrated in Table 4, exploring methods for our task can also lead to boosting performance in these settings.
> > >
> > > Finally, we note that we propose a task as well as a new framework that we use to adapt the methods proposed by prior work on our task. We find that the ranking of methods varies significantly when adapted to our task, identifying the need for direct exploration.  Also, our experiments cover adapting five methods and evaluate on three datasets containing real-world noise.  In contrast, most LNL papers propose a single method, and evaluate on only one or two real-world datasets.  Thus, our paper is more general, enabling us to generalize to datasets not supported by prior work as well as still benefiting their settings, while also providing more thorough experiments.

---

### Official Review · Reviewer_3qaZ · 2023-10-29

**Soundness:** 2 fair
**Presentation:** 2 fair
**Contribution:** 2 fair
**Rating:** 5
**Confidence:** 4

**Summary:**

The paper explores an overlooked task of learning with noisy labels utilizing noise source knowledge. The paper proposes a simple wrapper method that can be used on top of existing methods for noise labels. Overall, the paper studies an interesting setup and the proposed method is practical. However, there are some concerns on the applicability of the method and also the experiment evaluation.

**Strengths:**

1. Intuitively, using noise source knowledge can indeed be helpful.
2. The paper proposes a simple method that can be plugged into existing base models.
3. Experiments shows improvement from the proposed method.

**Weaknesses:**

1. Writing can be improved. There are simple errors that should have been avoided by proof reading. For example. “when considering the presence of the noise source yellow class, it becomes evident that these noisy samples are closer to their true label class.” Where is “yellow” class in the figure? This causes confusion as “yellow” class is also referred in Equation 2.
2. The applicability of the method should be made more clear and more intuition should be provided.
3. Experiment setup can be improved to better justify the claims.

**Questions:**

1. It seems the method works in the scenaio where there are confusing class pairs on which examples are miss labeled. However, in real-world, there can be many more noise label patterns in a same dataset, for example random white noise. How does the method work with the coexistence of other noise patterns?
2. It is mentioned that the proposed method "are effective on datasets where noise represents the majority of samples". Given a new dataset at hand with only noisy ground-truth labels, how should one decide whether to use the proposed method? and how would one know whether noise represents the majority of samples?
3. How often and also when is equation 2 different from selecting the highest probability class? Can you provide some numbers and examples from experiments? This can be helpful to understand the method better.
4. How is the model trained after detecting the noise examples? Do you just drop the noise examples?
5. It would be the best to also consider baselines that uses robust loss functions/regularization/etc

---

> ### Author Response · Authors · 2023-11-14
> **Reply (part 1)**
>
> Thank you for your efforts in reviewing our paper, and we will use your comments to improve our paper.  We have answered your questions below, and asked for additional clarifications for questions that were unclear to us.
>
> >W1. Writing can be improved. There are simple errors that should have been avoided by proof reading. For example. “when considering the presence of the noise source yellow class, it becomes evident that these noisy samples are closer to their true label class.” Where is “yellow” class in the figure? This causes confusion as “yellow” class is also referred in Equation 2.
>
> Apologies, the color change was a last-minute change to make them easier to see.  Where we refer to “yellow” it should be “blue” in the current figure.  We will use these comments to adjust our paper
>
> >W2. The applicability of the method should be made more clear and more intuition should be provided.
>
> >W3. Experiment setup can be improved to better justify the claims.
>
> We responded to your questions related to these comments below.  Please let us know if there are additional questions you have to address these comments.
>
> >Q1. It seems the method works in the scenaio where there are confusing class pairs on which examples are miss labeled. However, in real-world, there can be many more noise label patterns in a same dataset, for example random white noise. How does the method work with the coexistence of other noise patterns?
>
> Thank you for the opportunity to discuss this point! The existence of other source of noise (e.g., non-label noise) arises in both cell datasets we explore.  Specifically, each compound being tested on the cell datasets are collected over multiple experiments.  Basically- the same experiment is conducted multiple times, and there may be some small variations in exactly how each was performed.  In addition, there may also be differences in microscopes or the stain used to image the cells.  These differences are commonly referred to as “batch effects” and also make learning on these datasets challenging. Thus, our results on these cell datasets also demonstrate that we can achieve performance gains even when only some potential sources of noise (i.e., label noise) are known.
> >Q2. It is mentioned that the proposed method "are effective on datasets where noise represents the majority of samples". Given a new dataset at hand with only noisy ground-truth labels, how should one decide whether to use the proposed method? and how would one know whether noise represents the majority of samples?
>
> It's crucial to emphasize that our knowledge-integrated methods exhibit effectiveness not only in high noise ratios but also provide benefits across a broad spectrum of noise ratios. When dealing with a new dataset, if there exists available knowledge about the noise sources, such as information on confusing pairs in the metadata, it is worthwhile to consider employing LNL+K methods.
> >Q3. How often and also when is equation 2 different from selecting the highest probability class? Can you provide some numbers and examples from experiments? This can be helpful to understand the method better.
>
> Thank you for highlighting this. A potential experimental validation can be found by examining the precision and recall rates of clean the sample selection. We present statistics from CIFAR10 with a dominant noise ratio of 0.8 as an illustrative example:
>
> | Method  | precision | recall | classification accuracy |
> |---------|-----------|--------|-------------------------|
> | CRUST   | 72.29     | 36.15  | 65.79                   |
> | CRUST+k | 87.67     | 99.04  | 80.54                   |
> | FINE    | 88.53     | 61.89  | 75.45                   |
> | FINE+k  | 89.64     | 99.61  | 80.52                   |
> | SFT     | 97.27     | 94.37  | 75.43                   |
> | SFT+k   | 98.99     | 94.95  | 76.78                   |
>
> Clearly, both precision and recall rates for sample selection witness an increase with the adoption of the LNL+K method. The substantial improvement in recall rate underscores the superior ability of LNL+K in retaining hard negative samples for training.

---

> > ### Author Response · Authors · 2023-11-14
> > **Reply (part 2)**
> >
> > >Q4. How is the model trained after detecting the noise examples? Do you just drop the noise examples?
> >
> > The exact approach varies depending on the method used.  +K methods only adapt the sample detection part, and the rest of the training strategy remains the same as the original base method. For example,  CRUST is a sample selection method and it only trains with the selected clean sample subset. UNICON incorporates semi-supervised learning for noisy samples, after detection, it uses mix-up and pseudo labels for training all samples. Each of these methods has its own pros and cons, but many can be adapted to take advantage of noise source knowledge.
> >
> > >Q5. It would be the best to also consider baselines that uses robust loss functions/regularization/etc
> >
> > Thanks for the suggestion, we incorporated an extra baseline method, SOP[1], into the synthesized noisy dataset. The results indicate that our adaptation methods consistently surpass the performance of SOP.
> >
> > | Dataset | Noise Type | Noise Ratio | SOP Accuracy (%) | Best LNL+K Accuracy (%) |
> > |---------|------------|-------------|-------------------|--------------------------|
> > | CIFAR10 | asym       | 0.2         | 92.85 +- 0.49     | 93.19 +- 0.08            |
> > | CIFAR10 | asym       | 0.4         | 89.93 +- 0.25     | 91.51 +- 0.12            |
> > | CIFAR10 | dominant   | 0.2         | 89.86 +- 0.40     | 90.83 +- 0.11            |
> > | CIFAR10 | dominant   | 0.5         | 86.94 +- 0.37     | 89.21 +- 0.42            |
> > | CIFAR10 | dominant   | 0.8         | 80.65 +- 0.71     | 82.27 +- 0.29            |
> > | CIFAR100| asym       | 0.2         | 72.60 +- 0.70     | 76.87 +- 0.24            |
> > | CIFAR100| asym       | 0.4         | 70.58 +- 0.30     | 73.91 +- 0.11            |
> > | CIFAR100| dominant   | 0.2         | 62.47 +- 0.47     | 66.77 +- 0.54            |
> > | CIFAR100| dominant   | 0.5         | 55.78 +- 0.68     | 61.55 +- 0.13            |
> > | CIFAR100| dominant   | 0.8         | 45.94 +- 0.65     | 48.47 +- 0.40            |
> >
> >
> > [1]Liu, Sheng, et al. "Robust training under label noise by over-parameterization." ICML, 2022.

---

### Official Review · Reviewer_X4gK · 2023-10-30

**Soundness:** 2 fair
**Presentation:** 2 fair
**Contribution:** 1 poor
**Rating:** 3
**Confidence:** 5

**Summary:**

The work focuses on a very practical problem - learning with label noise. It introduces the concept of LNL+K, which incorporates additional 'noise source knowledge' into existing sample selection methods. Specifically, this work utilizes additional 'noise source knowledge' to identify potentially confusing noise classes $D_{c-ns}$. Instead of considering the probability of the annotated class being clean~($P_{c}$), the proposed approach compares the probability of a given label being clean with that of the confusing classes. The proposed strategy is combined with various existing sample selection methods and evaluated on several datasets.

**Strengths:**

1. The presentation is clear.
2. Multiple existing sample selection methods are considered for evaluation.

**Weaknesses:**

1. The novelty is somewhat limited. As quoted in the paper - "The selected sample’s probability may not be the highest", indeed current sample selection methods, such as the widely-applied GMM style, possibly lead to false positives (hard negatives), as they only consider whether the annotated label is 'clean enough or not' while ignoring its relative 'cleanness' versus other classes. Upon this, though introduced as 'noise source knowledge', the core idea of this work is to compare the 'cleanness' of the annotated class versus other classes, which is straightforward and trivial in existing sample selection heuristics, and been applied already ('consistency measure in [1], probablity difference in [2], there should be more work in major venues). This may sounds a bit stringent, but I expect more insights rather than rephrasement, especially for venues like ICLR.

2. I expect more 'real noise source knowledge' to be considered, rather than current ones (transition matrix, etc.), which still rely on the noisy labels and current in-training models, leading to self-confirmation again.

3. The confusion class set \(D_{c-ns}\) induced by 'noise source knowledge' involves new hyperparameters. More ablations are necessary.

4. For a sample selection strategy, there always exists a dilemma of precision and recall, especially when extra hyperparameters are involved. This requires more detailed analysis.

5. The considered real-world noisy datasets lack persuasiveness. Experiments should be conducted on at least Clothing1m and WebVision.

[1] SSR: An Efficient and Robust Framework for Learning with Unknown Label Noise, BMVC2022
[2] P-DIFF: Learning Classifier with Noisy Labels based on Probability Difference Distributions, ICPR2020

**Questions:**

See weakness.

---

> ### Author Response · Authors · 2023-11-14
> **Reply (part 1)**
>
> Thank you for your efforts in reviewing our paper, and we will use your comments to improve our paper.  We have answered your questions below.
>
> >1. The novelty is somewhat limited. The core idea of this work is to compare the 'cleanness' of the annotated class versus other classes, which is straightforward and trivial in existing sample selection heuristics, and been applied already ('consistency measure in [C1], probablity difference in [C2], there should be more work in major venues). This may sounds a bit stringent, but I expect more insights rather than rephrasement, especially for venues like ICLR.
>
> We will note that the consistency measure and probability differences in [C1] and [C2] are not consistent with our introduced noise source knowledge formulation, and thus, it is not a simple rephasement of these methods.  To illustrate this point, let us begin by defining these criteria:
>
> [C1] defines consistency measure as “a high consistency measure ci at a sample xi means that its neighbours agree with its current label l^r_i — this indicates that l^r_i is likely to be correct. By setting a threshold θs to ci, a clean subset (Xc, Yrc) can be extracted.” We note that this consistency measure is very similar to that used by CRUST, which uses gradients rather than predicted labels to measure similarity between samples (which is also outperformed by LNL+K).
>
> [C2] defines probability difference as “δ = py − pn where pn is the largest component except p.”
>
> Now let’s consider the three following samples from the same neighborhood (ie., they are the samples used by [C1]) with predicted class probabilities for classes [cat, dog, bird]:
>
> Sample A: [0.20, 0.10, 0.60] with given and true labels cat
>
> Sample B: [0.25, 0.15, 0.60] with given and true labels bird
>
> Sample C: [0.15, 0.15, 0.70] with given and true labels bird
>
> Now, when deciding whether to label Sample A as noisy, [C1] would refer to its neighbors, and since a majority of labels of its neighbors disagree, it would identify it as a noisy sample, but may retain Sample B and C depending on the exact threshold used.  [C2] would compare the differences in probabilities, and would result in the same selection as [C1] (identifying A as noise, but possibly retain B and C).  However, in LNL+K, if the noise sources for “cat” is only “dog” and not “bird, then we would retain all three samples as clean.  This is because of the fact that our criteria would effectively ignore the high-confidence prediction for bird made for Sample A because we know that a cat would not be mislabeled as a bird sample.  Instead, it indicates this is likely a hard negative, perhaps due to similar backgrounds (e.g., many bird images may be in wooded areas, and this particular cat sample may also be taken in a wooded area).
>
> While we agree that this premise is a simple idea, we note that the majority of work in LNL either ignores the noise sources entirely (such as [C1] and C[2]), leading to failures like the one above, or focuses on trying to estimate the noise.  This illustrates that this idea is only obvious in hindsight.  In addition, we note that noise estimation methods are not beneficial for datasets where the noise is always given, such as in the tasks addressed in CHAMMI and BBBC036- they effectively attempt to just predict a known entity.  Also, the focus of noise estimation methods is typically, as the name suggests, on how to more effectively estimate the noise, but not necessarily on good strategies to use the noise source knowledge once it is estimated.
>
> In fact, an inherent issue with studying methods on how to effectively use the noise when exploring noise estimation methods is that it conflates errors due to issues with noise estimation with effective use of these methods.  In other words, some methods may be able to effectively use noise sources, but only when those are accurate representations of the data, and may underperform when the estimates are noisy.  This is where our work provides a contribution, as it can fairly evaluate such methods.  Our work also highlights the need for determining how to more effectively use noise sources either when they are known, such as for CHAMMI, BBBC036, and Animal10N.  Finally, we also demonstrate that our work can be beneficial when the noise sources are not known and instead have to be estimated, as shown in our experiments in Table 4.  Thus, our work provides novel insights into the LNL problem not found in prior work.
>
> [C1] SSR: An Efficient and Robust Framework for Learning with Unknown Label Noise, BMVC2022
>
> [C2] P-DIFF: Learning Classifier with Noisy Labels based on Probability Difference Distributions, ICPR2020

---

> > ### Author Response · Authors · 2023-11-14
> > **Reply (part 2)**
> >
> > >2. I expect more 'real noise source knowledge' to be considered, rather than current ones (transition matrix, etc.), which still rely on the noisy labels and current in-training models, leading to self-confirmation again.
> >
> > We note that we perform experiments on three different real source knowledge benchmarks, that represent five different applications and imaging settings.  CHAMMI itself contains three different datasets and is designed to test many different generalization properties of models, and BBBC036 and Animal10N expand on this.  This already represents a significant increase in the real noise source knowledge settings explored in prior work that typically only consider one or two real-noise source knowledge datasets  (e.g., [1,2])  rather than the five settings explored in prior work.
> >
> > [1] Yi, Li, et al. "On learning contrastive representations for learning with noisy labels." CVPR. 2022.
> >
> > [2] Qi Wei, Haoliang Sun, Xiankai Lu, and Yilong Yin. Self-filtering: A noise-aware sample selection for label noise with confidence penalization. ECCV, 2022.
> > >3. The confusion class set (D_{c-ns}) induced by 'noise source knowledge' involves new hyperparameters. More ablations are necessary.
> >
> > Unfortunately, we are uncertain what “new hyperparameters” you are referring to, as the confusion class set is not a hyperparameter.  Rather, it is a given label in the meta data.  For example, in the BBBC036 dataset the set of ns classes are those that were explicitly designed by the scientists creating the dataset as a “do nothing” category.  For Animal10N, they are the set of categories that were explicitly noted by the authors that were often confused with each other.
> >
> > >4. For a sample selection strategy, there always exists a dilemma of precision and recall, especially when extra hyperparameters are involved. This requires more detailed analysis.
> >
> > Note that our work does not introduce any new hyperparameters.  Instead, we use the hyperparameters of the various benchmarks according to the guidelines presented by the authors.  If the reviewer wishes for a specific experiment, we would be happy to provide it to them.
> >
> > >5. The considered real-world noisy datasets lack persuasiveness. Experiments should be conducted on at least Clothing1m and WebVision.
> >
> > We begin by noting that the reviewer provides no justification for their assertions that our datasets are not persuasive, and, thus, provides no insight into what they feel is not being evaluated.  In fact, we note that our benchmarks are more general and test more settings with a greater potential societal impact than those suggested by the reviewer, and, thus, provide more evidence that our paper provides a meaningful contribution.
> >
> > In their comment, the reviewer also seems to suggest that exploring Clothing1M and WebVision has greater potential impact, but we strongly disagree with these sentiments.  Let us consider different axes by which we could consider how well Clothing1M and WebVision generalize.  In terms of image distribution, In addition, Clothing1M and WebVision are limited in that they use the same type of imaging (RGB), represent only two domains of data, and also only contain internet collected samples that itself represent very biased datasets as we will expand on later.  In contrast, we explore three different datasets of varying imaging types, RGB images in Animals10N, which cover similar applications as the datasets suggested by the reviewer, as well as cell microscopy images. Additionally, CHAMMI-CP is a combination of samples from two different experiment types, one measuring the response of cells to different drug treatments and other testing gene over-expression experiments, which is important for evaluating drug diversity projects. This already demonstrates that our study is already far more general than those suggested by the reviewer as they cover a broader range of tasks, applications, domains, and imaging types.
> >
> > In terms of potential impact of boosting performance on these datasets, we argue our benchmarks are also more favorable.   Not only do we also explore RGB images like Clothing1M and WebVision, but we also generalize to cell data, which understanding and analysis of this type of data is an entire field of study on its own.  In addition, as both suggested datasets are internet collected images, they overrepresent wealthy countries, and as such it is these wealthier countries benefit the most from addressing problems related to e-commerce (i.e., Clothing1M).  In contrast, the cell datasets represent a variety of applications and generalization settings, which can directly affect healthcare. As such, both wealthy and poor countries could benefit from this work. Thus, we argue that the datasets already explored in our paper represent greater potential for impact than those suggested by the reviewer, and are clearly well above the bar for publication.

---

### Official Review · Reviewer_kCaM · 2023-10-31

**Soundness:** 3 good
**Presentation:** 2 fair
**Contribution:** 3 good
**Rating:** 5
**Confidence:** 5

**Summary:**

This paper proposed incorporating noise source knowledge into some sample selection methods by comparing the confidence of noisy labels and noise source label. The proposed method is simple and easy to be integrated into multiple existing LNL methods. Experiments confirm the effectiveness of the proposed method in certain cases.

**Strengths:**

1. The studied problem that how to exploit noise source knowledge in noisy label learning is novel and interesting.
2. The proposed method is very simple but reasonable, which can be integrated into many methods.
3. The datasets include cell datasets which shows the potential of the proposed method in scientific research.

**Weaknesses:**

1. As shown in the experiments, in some cases, the proposed method will lead to a decrease in performance. The authors should offer a deeper analysis about the reason of the decrease in performance and when could the performance gain be ensured.
2. The real-world datasets are small-scale and special. It would be better to test the performance of the proposed methods with estimated noise source knowledge in more large and general benchmarks, e.g., Clothing1M and WebVision.
3. Some writing need further clarification. First, how to generate dominant noise is still unclear.  Are there multiple noise sources for each recessive label. And why is the dataset still balanced after mislabeling in these cases? It seems that the number of examples labeled by dominant classes is less than recessive classes. Second, how to use DualT to estimate noise source knowledge is not clear.
4. (Minor) The related works can include more recent classifier-consistent methods, e.g. [1,2,3].
5. (Minor) What does the "noise supervision" mean?

[1] Estimating Noise Transition Matrix with Label Correlations for Noisy Multi-Label Learning. NeurIPS 2022

[2] A holistic view of noise transition matrix in deep learning and beyond. ICLR 2023

[3] Identifiability of label noise transition matrix. ICML 2023

**Questions:**

See above weaknesses.  I am happy to increase my score if my concerns are addressed.

---

> ### Author Response · Authors · 2023-11-14
> **Reply (part 1)**
>
> Thank you for your efforts in reviewing our paper, and we will use your comments to improve our paper.  We have answered your questions below.
>
> >1. As shown in the experiments, in some cases, the proposed method will lead to a decrease in performance. The authors should offer a deeper analysis about the reason of the decrease in performance and when could the performance gain be ensured.
>
> Thank you for your attention to these results. As discussed in section 4.3, we term the various performance improvements as the "knowledge absorption rate," a metric relevant to noise type, noise ratio, and the base LNL methods. Among all our findings, only a few exhibited a performance decrease (8 settings out of 52), and the majority of those (5 settings out of 8) decreased by less than 0.5%. The decline in CRUST^{+k} performance is likely attributable to the need for careful tuning of a hyperparameter controlling the size of the clean sample selection, especially in the case of CRUST^{+k}. SFT^{+k} encounters challenges at exceptionally high noise ratios, suggesting that knowledge about the noise source is particularly beneficial for feature-based clean sample detection methods in such conditions.
>
> >2. The real-world datasets are small-scale and special. It would be better to test the performance of the proposed methods with estimated noise source knowledge in more large and general benchmarks, e.g., Clothing1M and WebVision.
>
> We will note that the real world datasets are evaluated on are of similar sizes as other common fine-tuning benchmarks.  For example, BBBC036 contains 156K images, making it on par with datasets like COCO and Visual Genome.  We note that by definition, the task we are exploring is one that requires labeled samples.  Labels are expensive, and as such, collecting very large datasets can be prohibitively expensive for many applications.  This is especially true in scientific research settings like those analyzing the cell datasets where LNL+K is a natural fit as these datasets can require significant and very expensive expertise to annotate. Thus, settings where data and labels are *very* plentiful, like clothing Clothing1M and WebVision, actually represent very narrow application areas of LNL and related tasks.
>
> More specifically, let us consider different axes by which we could consider how well Clothing1M and WebVision generalize.  In terms of image distribution, In addition, Clothing1M and WebVision are limited in that they use the same type of imaging (RGB), represent only two domains of data, and also only contain internet-collected samples that represent very biased datasets as we will expand on later.  In contrast, we explore three different datasets of varying imaging types, RGB images in Animals10N, which cover similar applications as the datasets suggested by the reviewer, as well as cell microscopy images. Additionally, CHAMMI-CP is a combination of samples from two different experiment types, one measuring the response of cells to different drug treatments and the other testing gene over-expression experiments, which is important for evaluating drug diversity projects. This already demonstrates that our study is far more general than those suggested by the reviewer and used in prior work (e.g., [1,2]) as they cover a broader range of tasks, applications, domains, and imaging types.
>
> In their comment, the reviewer also seems to suggest that exploring Clothing1M and WebVision has greater potential impact, but we strongly disagree with these sentiments.  We generalize to cell data, which understanding and analysis of this type of data is an entire field of study on its own.  Thus,  exploring methods that can benefit these applications has far more potential for high impact than simply boosting performance on Clothing1M and WebVision. The cell datasets represent a variety of applications and generalization settings, including drug discovery.  In other words- these datasets represent tasks that can directly affect healthcare. Although one could argue that RGB images contain an important domain, we do already explore this setting with the Animals10N dataset.  Thus, we argue that the datasets already explored in our paper represent greater potential for impact than those suggested by the reviewer, and are clearly well above the bar for publication.
>
> [1] Yi, Li, et al. "On learning contrastive representations for learning with noisy labels." CVPR. 2022.
>
> [2] Qi Wei, Haoliang Sun, Xiankai Lu, and Yilong Yin. Self-filtering: A noise-aware sample selection for label noise with confidence penalization. ECCV, 2022.

---

> > ### Author Response · Authors · 2023-11-14
> > **Reply (part2)**
> >
> > >3. Some writing need further clarification. First, how to generate dominant noise is still unclear. Are there multiple noise sources for each recessive label. And why is the dataset still balanced after mislabeling in these cases? It seems that the number of examples labeled by dominant classes is less than recessive classes.
> >
> > Thanks for the suggestion and we’ll improve the writing in the revision. Additional information regarding the primary noise generation is available in Appendix B. In each recessive class, there are multiple noise sources, with all dominant classes serving as the noise sources. To illustrate, in CIFAR-10, classes 6-10 are considered recessive, and instances of these classes might be incorrectly labeled as dominant classes 1-5. We provide a breakdown of the noise composition in Table 5. To maintain balance after mislabeling, we adopt an unbalanced sampling approach to construct the dataset. For instance, with a noise ratio of 0.5 in the CIFAR-10 dataset, we sample 1250 instances for each dominant class and 3750 instances for each recessive class. After mislabeling 1250 samples to the dominant class for each recessive class, there are 2500 samples in each class.
> > >Second, how to use DualT to estimate noise source knowledge is not clear.
> >
> > Thank you for bringing it up. Exploring classifier-consistent methods for estimating noise sources is a promising new avenue for research. In this paper, we've taken an initial step by employing a straightforward approach: obtaining the noise transition matrix from DualT. For each class, we designate the class with the second-largest transition probability as the noise source. For instance, if P(cat | cat) = 0.6 and P(dog | cat) = 0.3, we identify "cat" as the noise source for "dog" because images labeled as "dog" could arise from the true label "cat." It's important to note that we use DualT as an illustrative example to demonstrate the potential of bridging classifier-consistent and inconsistent methods. This connection holds significance in the feature research within the LNL field, as showcased in this paper.
> >
> > >4. (Minor) The related works can include more recent classifier-consistent methods, e.g. [1,2,3].
> >
> > Thanks for the suggestions! We’ll add these in the revision.
> >
> > >5. (Minor) What does the "noise supervision" mean?
> >
> > We refer to “knowing the sources of the noise” as a form of "supervision" as it adds a signal about what the noise may look like.

---

### Official Review · Reviewer_unqu · 2023-10-31

**Soundness:** 1 poor
**Presentation:** 1 poor
**Contribution:** 2 fair
**Rating:** 3
**Confidence:** 2

**Summary:**

The authors propose a new task and method for the task of Noise Source Knowledge Integration. They assume that a set of possibly confusing classes for a given other class is made available e.g. the knowledge that trucks and automobiles are easily confused with each other. They integrate this knowledge in the learning process and demonstrate improved results on synthetic and real world datasets when their method is combined with various selected methods from the noisy labels literature which currently do not integrate noise source knowledge.

**Strengths:**

- The authors report broadly positive results with improvements on most datasets in most settings.

- The authors demonstrate that their method can be successfully combined with a range noisy labels learning methods.

- The use of the CIFAR-10 and CIFAR-100 synthetic noise datasets is relatively standard in the noisy labels literature, and the authors introduce some non-standard evaluation datasets with interesting scientific applications (BBBC036, CHAMMI-CP).

**Weaknesses:**

- I found the presentation extremely hard to follow, with many terms' definitions unclear to me and/or errors in the notation and presentation that made following the paper very difficult. I have listed what I regard as errors below and have deferred to the questions section various unclear terms for clarification. I am open to substantially improving my score if it is clear I have misunderstood the paper and it is clarified to me, but at present given my careful reading of the presentation I find the entire method definition and hence its evaluation unconvincing as I cannot understand it.
    - There are repeated references to a yellow class in Figure 1 e.g. "the noise source yellow class", "p(yellow | x_i) > p(red ? x_i)", there is no yellow class as far as I can see.
    - The output of algorithm 1 is denoted P(X) = {p(\tilde{y}_i | x_i)}, this neither seems to be the correct definition of P(X) or the algorithm output.


Small points:
- $\tilde{Y}$ is used in the literature and in section 2 to signify noisy labels, yet in paragraph 2 of section 3, this notation is flipped and now $\tilde{y}$ are the "true labels".

**Questions:**

- The definition of noise sources is unclear to me. Can you please confirm if my understanding based on the definition at the end of the second paragraph in section 3 is correct: noise sources for a given class c, is the set of all other classes such that there is a non-zero probability of a data point that is truly in class c being mislabeled as that class?
    - If my understanding is correct, then I find it hard to understand how this information is useful to the extent demonstrated in the experiments. Firstly, strictly based on this definition, all classes should be in the noise source set as there is a non-zero probability that a class c is mislabelled to any other class. However it seems to be the case that the definition of this set is in fact more loosely applied in the experimental section of the paper where the noise sources set for class c are really classes that have a relatively high probability of being confused for class c. Nonetheless I struggle to see how this information would be as useful as some of the experimental results seem to show, as essentially this additional information would merely say on a dataset level which classes are reasonably likely to be confused for each other. This is not per sample/input dependent nor does it convey as much information as the true transition matrix which would contain transition probabilities and not merely binary values for each class pair. Could the authors please comment?

- Figure 1 is very unclear to me, based on my understanding of noise sources above, then noise sources are not data points but classes, yet Figure 1 presents new data points as noise sources? In addition could the authors please clarify the meaning of red, blue, circles and triangles in the figure. For example, what is the meaning of a blue triangle?

- I do not understand the arguments leading to equation 2 in section 3.1. In particular:
    - "Fig. 1 has a high probability of belonging to the red class, i.e., p(red|xi) > δ, then it is detected as a clean sample in LNL. However, compared to the probability of belonging to the noise source yellow class, p(yellow|xi) > p(red|xi), so the red triangle is detected as a noisy sample in LNL+K." As in this paper, noisy labels methods are usually evaluated on multi-class classification tasks. Hence I do not understand how a standard LNL method would fail in this case when p(yellow|xi) > p(red|xi), then while p(red|xi) > δ it must be the case that p(yellow|xi) >> δ and hence the sample would be labelled as yellow by the standard LNL method. Fundamentally I fail to see how standard LNL methods would not satisfy equation 2, please explain?

- How the method is incorporated into the various LNL methods is unclear to me. I can understand how the various methods currently identify clean labels. But a very short description is given of the integration in each of the 5 cases. I do not think the level of detail would be sufficient to replicate your results or to combine the method with another LNL method. For example: "To estimate the likelihood of a sample label being clean in CRUST+k, we mix this sample with all other noise source class samples and apply CRUST to the combined set. If the sample is selected as part of the noise source class cluster, we assume its label is noisy." What does it mean to mix the "samples"? What are the samples in this case? The noise sources as defined above are simply a set of classes, they are not input dependent nor can I see how they can be mixed with training examples?

---

> ### Author Response · Authors · 2023-11-14
> **Reply (part 1)**
>
> Thank you for your efforts in reviewing our paper, and we will use your comments to improve our paper.  We have answered your questions below, and asked for additional clarifications for questions that were unclear to us.
>
> >There are repeated references to a yellow class in Figure 1 e.g. "the noise source yellow class", "p(yellow | x_i) > p(red | x_i)", there is no yellow class as far as I can see.
>
> Apologies for the confusion, where “yellow” is referenced it should be “blue.” We have adjusted our discussions accordingly.
>
> >The output of algorithm 1 is denoted P(X) = {p(\tilde{y}_i | x_i)}, this neither seems to be the correct definition of P(X) or the algorithm output.
>
> The algorithm output P is a list of length n, where n is the number of samples in the inputs X. P[i] denotes the probability of sample $x_i$ being clean, i.e. p(\widetilde{y_i}|x_i). This comment is confusing to us, we have defined P(X) ourselves in the algorithm, P(X) = {p(\widetilde{y_i}|x_i)}_{i=1}^n, so it can only be “correct” according to our definition.  As such,  we would appreciate some clarity on your comment.
>
>
> >Y is used in the literature and in section 2 to signify noisy labels, yet in paragraph 2 of section 3, this notation is flipped and now y are the "true labels"
>
> Thank you for your comment, we will ensure that the notation is consistent.
> >The definition of noise sources is unclear to me. Can you please confirm if my understanding based on the definition at the end of the second paragraph in section 3 is correct: noise sources for a given class c, is the set of all other classes such that there is a non-zero probability of a data point that is truly in class c being mislabeled as that class?
>
> To clarify, noise sources are defined on a category level.  So let’s say for category *c* we are told its noise sources are categories *d* and *e*, it means that for any image for category *c* that is mislabeled, we know that its true label must either be an image category *d* or an image of category *e*.
>
> > Firstly, strictly based on this definition, all classes should be in the noise source set as there is a non-zero probability that a class c is mislabelled to any other class. However it seems to be the case that the definition of this set is in fact more loosely applied in the experimental section of the paper where the noise sources set for class c are really classes that have a relatively high probability of being confused for class c. Nonetheless I struggle to see how this information would be as useful as some of the experimental results seem to show, as essentially this additional information would merely say on a dataset level which classes are reasonably likely to be confused for each other. This is not per sample/input dependent nor does it convey as much information as the true transition matrix which would contain transition probabilities and not merely binary values for each class pair. Could the authors please comment?
>
> Our understanding of this comment is that the reviewer wishes to know 1. The identification of noise sources, 2. Knowing why categories that may often be confused with each other would be useful, as it conveys information at a dataset level and not on a per-sample level.
>
> For 1, in theory, we define those noise sources with “significantly large” non-zero probabilities. In practice, for the real-world dataset, the noise source information is collected from the dataset metadata, e.g. confusing pairs from Animal10N dataset and control group in cell datasets.
>
> Regarding 2, we note that two images may be considered hard-negatives with each other not because they contain the same category, but because there are other correlated attributes.  For example, let’s “cat” image x_1 is in a wooded environment and the dataset also contains many “bird” images that are also in similar wooded environments.  As such, although a human annotator may not confuse the cat for a bird, i.e., they are not noise sources for each other, a neural network may find them more challenging to distinguish between each other as the background would be the same.  Prior work may disregard these samples as they are not confident in their true label due to the high confidence prediction in “bird.”  However, in our work, we would retain all such samples.  Thus, by considering such information, we would retain more hard-negative samples than prior work.

---

> ### Author Response · Authors · 2023-11-14
> **Reply (part 2)**
>
> >Figure 1 is very unclear to me, based on my understanding of noise sources above, then noise sources are not data points but classes, yet Figure 1 presents new data points as noise sources?
>
> Noise sources are retained from classes, however, Figure 1 represents how data selection methods may differ in our setting and prior work.  In the example given, we show that in cases where many samples may be mislabeled, then the model may begin to model the noise as the true class.  However, in our work, any sample that follows the same distribution as the noise sources would be removed (represented as the noise source data points).  We will clarify this point in a new figure.
> >In addition could the authors please clarify the meaning of red, blue, circles and triangles in the figure. For example, what is the meaning of a blue triangle?
>
> These are defined at the bottom of the figure.  Colors refer to the noisy labels, whereas shapes refer to the true labels. Red circles and blue triangles are clean samples, and red triangles are noisy samples whose labels are red but true labels are blue.
> >"Fig. 1 has a high probability of belonging to the red class, i.e., p(red|xi) > δ, then it is detected as a clean sample in LNL. However, compared to the probability of belonging to the noise source blue class, p(blue|xi) > p(red|xi), so the red triangle is detected as a noisy sample in LNL+K." As in this paper, noisy labels methods are usually evaluated on multi-class classification tasks. Hence I do not understand how a standard LNL method would fail in this case when p(blue|xi) > p(red|xi), then while p(red|xi) > δ it must be the case that p(blue|xi) >> δ and hence the sample would be labelled as blue by the standard LNL method. Fundamentally I fail to see how standard LNL methods would not satisfy equation 2, please explain?
>
> Note that as mentioned in Figure 1 and reiterated above, colors are noisy labels, whereas shapes are true labels.  In other words, red circles are correctly labeled, whereas red triangles are incorrectly labeled.  That said, to answer your question, a fundamental difference between LNL methods and LNL+K is that they use different criteria to select samples.  **For an LNL method, it does not matter if a sample would satisfy Eq. 2 since this *is not the LNL criteria.***  Instead, LNL methods use the criteria p(red|xi) > δ.   In other words, LNL methods often ask the question, what samples am I most confident are of the correct class (red in this case)?  In contrast, LNL+K asks, what samples am I most confident are not noise? These may seem like a subtle difference, but in practice it can have significant ramifications as demonstrated in our experiments. Effectively, the LNL criteria only requires that samples have a high likelihood of being a clean label *relative to other samples*, but all samples may have *low absolute likelihood* of being a clean label.  As such, LNL+K criteria is more suitable, as it does not measure the likelihood of being clean, but rather likelihood of not being noise.
>
> >How the method is incorporated into the various LNL methods is unclear to me. I can understand how the various methods currently identify clean labels. But a very short description is given of the integration in each of the 5 cases. I do not think the level of detail would be sufficient to replicate your results or to combine the method with another LNL method.
>
> We apologize for the confusion.  As noted in Section 3.2, we provide a high-level summary of each method in the main paper, but a detailed description of how we modify each method for LNL+K is provided in Appendix A. We shall clarify these methods in a new version.
>
> > What does it mean to mix the "samples" in CRUST+k? What are the samples in this case?
>
> The answers for these questions can be inferred from the earlier portion of this description in Section 3.2. I.e.,CRUST works by performing pairwise gradient distance within a class for clean sample selection, and selects those with the most similar gradients.  Thus, the “samples” are the data points that are labeled for a category (e.g., the red data points in Figure 1), and mixing them means that we would add the data samples for the noise source to this set (which would be the blue samples in Figure 1b).  This is further discussed in Appendix A.
>
> >The noise sources as defined above are simply a set of classes, they are not input dependent nor can I see how they can be mixed with training examples?
>
> The noise sources are a set of classes, but each class must have their own labeled samples.  Thus, we can mix the samples of these categories together.  In other words, if “dog” is given as a noise source for “cat” images, then we must have labeled images of “dogs” in the training set for this noise source to be informative (i.e., we have to have some way of understanding what a “dog” is).